# SPHERICAL TREE-SLICED WASSERSTEIN DISTANCE

**Viet-Hoang Tran**\*
Department of Mathematics
National University of Singapore
`hoang.tranviet@u.nus.edu`

**Thanh T. Chu**\*
Department of Computer Science
National University of Singapore
`thanh.chu@u.nus.edu`

**Khoi N.M. Nguyen**
FPT Software AI Center
`khoinnm1@fpt.com`

**Trang Pham**
Qualcomm AI Research$^\diamond$
`tranpham@qti.qualcomm.com`

**Tam Le**$^\dagger$
The Institute of Statistical Mathematics
& RIKEN AIP
`tam@ism.ac.jp`

**Tan M. Nguyen**$^\dagger$
Department of Mathematics
National University of Singapore
`tanmn@nus.edu.sg`

## ABSTRACT

Sliced Optimal Transport (OT) simplifies the OT problem in high-dimensional spaces by projecting supports of input measures onto one-dimensional lines and then exploiting the closed-form expression of the univariate OT to reduce the computational burden of OT. Recently, the Tree-Sliced method has been introduced to replace these lines with more intricate structures, known as tree systems. This approach enhances the ability to capture topological information of integration domains in Sliced OT while maintaining low computational cost. Inspired by this approach, in this paper, we present an adaptation of tree systems on OT problems for measures supported on a sphere. As a counterpart to the Radon transform variant on tree systems, we propose a novel spherical Radon transform with a new integration domain called spherical trees. By leveraging this transform and exploiting the spherical tree structures, we derive closed-form expressions for OT problems on the sphere. Consequently, we obtain an efficient metric for measures on the sphere, named Spherical Tree-Sliced Wasserstein (STSW) distance. We provide an extensive theoretical analysis to demonstrate the topology of spherical trees and the well-definedness and injectivity of our Radon transform variant, which leads to an orthogonally invariant distance between spherical measures. Finally, we conduct a wide range of numerical experiments, including gradient flows and self-supervised learning, to assess the performance of our proposed metric, comparing it to recent benchmarks. The code is publicly available at https://github.com/lilythchu/STSW.git.

## 1 INTRODUCTION

Despite being embedded in high dimensional Euclidean spaces, in practice, data often reside on low dimensional manifolds (Fefferman et al., 2016). The hypersphere is one such manifold with various practical applications. The range of applications involving distributions on a hypersphere is remarkably broad, underscoring the significance of spherical geometries across multiple fields. These applications encompass spherical statistics (Jammalamadaka, 2001; Mardia & Jupp, 2009; Ley & Verdebout, 2017; Pewsey & García-Portugués, 2021), geophysical data (Di Marzio et al., 2014), cosmology (Jupp, 1995; Cabella & Marinucci, 2009; Perraudin et al., 2019), texture mapping (Elad et al., 2005; Dominitz & Tannenbaum, 2009), magnetoencephalography imaging (Vrba & Robinson, 2001), spherical image representations (Coors et al., 2018; Jiang et al., 2024), omnidi-

---

\* Co-first authors. $^\dagger$ Co-last authors. $^\diamond$ Qualcomm Vietnam Company Limited. Correspondence to: hoang.tranviet@u.nus.edu & tanmn@nus.edu.sg

rectional images (Khasanova & Frossard, 2017), and deep latent representation learning (Wu et al., 2018; Chen et al., 2020; Wang & Isola, 2020; Grill et al., 2020; Caron et al., 2020; Davidson et al., 2018; Liu et al., 2017; Yi & Liu, 2023).

Optimal Transport (OT) (Villani, 2008; Peyré et al., 2019) is a geometrically natural metric for comparing probability distributions, and it has received significant attention in machine learning in recent years. However, OT faces a significant computational challenge due to its supercubic complexity in relation to the number of supports in input measures (Peyré et al., 2019). To alleviate this issue, several variants have been developed to reduce the computational burden, including entropic regularization (Cuturi, 2013), minibatch OT (Fatras et al., 2019), low-rank approaches (Forrow et al., 2019; Altschuler et al., 2019; Scetbon et al., 2021), the Sliced-Wasserstein distance (Rabin et al., 2011; Bonneel et al., 2015), tree-sliced-Wasserstein distance (Indyk & Thaper, 2003; Le et al., 2019; Le & Nguyen, 2021; Tran et al., 2025b;c;d), and Sobolev transport (Le et al., 2022; 2023; 2024).

**Related work.** There has been growing interest in utilizing OT to compare spherical probability measures (Cui et al., 2019; Hamfeldt & Turnquist, 2022). To mitigate the computational burden, recent studies have focused on sliced spherical OT (Quellmalz et al., 2023; Bonet et al., 2022; Tran et al., 2024b). Quellmalz et al. (2023) introduced the vertical slice transform and a normalized version of the semicircle transform to define sliced OT on the sphere. The semicircle transform was also employed in (Bonet et al., 2022) to define a spherical sliced Wasserstein. Meanwhile, Tran et al. (2024b) utilized stereographic projection to create a spherical distance between measures via univariate OT problems. However, projecting spherical measures onto a line or circle poses challenges due to the loss of topological information. Furthermore, comparing one-dimensional measures on circles is computationally more expensive, as it requires an additional binary search.

Notably, Tran et al. (2025b;d) offer an alternative method by substituting one-dimensional lines in the Sliced Wasserstein framework with more complex domains, referred to as tree systems. These systems operate similarly to lines but with a more advanced and intricate structure. This approach is expected to enhance the capture of topological information while preserving the computational efficiency of one-dimensional OT problems. Inspired by this observation, we propose an adaptation of tree systems to the hypersphere, called spherical trees, to develop a new metric for measures on the hypersphere. Spherical trees satisfy *two important criteria*: (i) spherical measures can be projected onto spherical trees in a meaningful manner, and (ii) OT problems on spherical trees admit a closed-form expression for a fast computation.

**Contribution.** Our contributions are three-fold:

1. We provide a comprehensive theoretical construction of spherical trees on the sphere, analogous to the notion of tree systems. We demonstrate that spherical trees, as topological spaces, are metric spaces defined by tree metrics, which ensures that OT problems on these spaces can be analytically solved with closed-form solutions.

2. We propose the Spherical Radon Transform on Spherical Trees, which transforms functions on the sphere to functions on spherical trees. We also present the concept of splitting maps for the sphere, a key component of this new Spherical Radon Transform, which describes how mass at a point is distributed across the spherical tree. In addition, we examine the orthogonal invariance of splitting maps, which later proves to be a sufficient condition for the injectivity of the Spherical Radon Transform.

3. We propose the novel Spherical Tree-Sliced Wasserstein (STSW) distance for probability distributions on the sphere. By selecting orthogonal invariant splitting maps, we demonstrate that STSW is a invariant metric under orthogonal transformations. Finally, we derive a closed-form approximation for STSW, enabling an efficient and highly parallelizable implementation.

**Organization.** The rest of the paper is organized as follows: we review Wasserstein distance variants in §2. We propose the notion of Spherical Trees on the Sphere with a formal construction in §3. We introduce the Spherical Radon Transform on Spherical Trees, and discusses its injectivity in §4. In §5, we propose Spherical Tree-Sliced Wasserstein (STSW) distance and derive a closed-form approximation for STSW. Finally, we evaluate STSW on various tasks in §6. Theoretical proofs and experimental details are provided in Appendix.

## 2   PRELIMINARIES

In this section, we review Wasserstein distance, Sliced Wasserstein distance, Wasserstein distance on tree metric spaces and Tree-Sliced Wasserstein distance on Systems of Lines.

**Wasserstein Distance.**   Let $\Omega$ be a measurable space, endowed with a metric $d$, and let $\mu$, $\nu$ be two probability distributions on $\Omega$. Denote $\mathcal{P}(\mu, \nu)$ as the set of probability distributions $\pi$ on the product space $\Omega \times \Omega$ such that $\pi(A \times \Omega) = \mu(A)$ and $\pi(\Omega \times B) = \nu(B)$ for all measurable sets $A$, $B$. For $p \geqslant 1$, the $p$-Wasserstein distance $W_p$ (Villani, 2008) between $\mu$, $\nu$ is defined as:

$$W_p(\mu, \nu) = \inf_{\pi \in \mathcal{P}(\mu,\nu)} \left( \int_{\Omega \times \Omega} d(x,y)^p \, d\pi(x,y) \right)^{\frac{1}{p}}. \tag{1}$$

**Sliced Wasserstein Distance.**   The Radon Transform (Helgason, 2011) is the operator $\mathcal{R} : L^1(\mathbb{R}^d) \to L^1(\mathbb{R} \times \mathbb{S}^{d-1})$ defined by: for $f \in L^1(\mathbb{R}^d)$, we have $\mathcal{R}f \in L^1(\mathbb{R} \times \mathbb{S}^{d-1})$ such that $\mathcal{R}f(t, \theta) = \int_{\mathbb{R}^d} f(x) \cdot \delta(t - \langle x, \theta \rangle) \, dx$. Note that $\mathcal{R}$ is a bijection. The Sliced $p$-Wasserstein (SW) distance (Bonneel et al., 2015) between $\mu, \nu \in \mathcal{P}(\mathbb{R}^d)$ is defined by:

$$SW_p(\mu, \nu) := \left( \int_{\mathbb{S}^{d-1}} W_p^p(\mathcal{R}f_\mu(\cdot, \theta), \mathcal{R}f_\nu(\cdot, \theta)) \, d\sigma(\theta) \right)^{\frac{1}{p}}, \tag{2}$$

where $\sigma = \mathcal{U}(\mathbb{S}^{d-1})$ is the uniform distribution on $\mathbb{S}^{d-1}$; and $f_\mu, f_\nu$ are the probability density functions of $\mu, \nu$, respectively.

**Tree Wasserstein Distances.**   Let $\mathcal{T}$ be a rooted tree (as a graph) with non-negative edge lengths, and the ground metric $d_\mathcal{T}$, i.e. the length of the unique path between two nodes. Given two probability distributions $\mu$ and $\nu$ supported on nodes of $\mathcal{T}$, the Wasserstein distance with ground metric $d_\mathcal{T}$, i.e., tree-Wasserstein (TW) (Le et al., 2019), yields a closed-form expression as follows:

$$W_{d_\mathcal{T}, 1}(\mu, \nu) = \sum_{e \in \mathcal{T}} w_e \cdot \left| \mu(\Gamma(v_e)) - \nu(\Gamma(v_e)) \right|, \tag{3}$$

where $v_e$ is the endpoint of edge $e$ that is farther away from the tree root, $\Gamma(v_e)$ is a subtree of $\mathbb{T}$ rooted at $v_e$, and $w_e$ is the length of $e$.

**Tree-Sliced Wasserstein Distances on Systems of Lines.**   Tree systems (Tran et al., 2025d) are proposed as replacements of directions in SW. As a topological space, they are constructed by joining (gluing) multiple copies of $\mathbb{R}$ based on a tree (graph) framework, forming a measure metric space in which optimal transport problems admit closed-form solutions. By developing a variant of the Radon Transform that transforms functions on $\mathbb{R}^d$ to functions on tree systems, Tree-Sliced Wasserstein Distances on Systems of Lines (TSW-SL) is are introduced in a similar manner as SW. The mentioned closed-form expressions lead to a highly parallelizable implementation for TSW-SL. We next extend the tree systems for measures on a sphere.

## 3   SPHERICAL TREES ON THE SPHERE

Let $d$ be a positive integer. Recall the notion of the $d$-dimensional sphere in $\mathbb{R}^{d+1}$,

$$\mathbb{S}^d := \left\{ x = (x_0, x_1, \ldots, x_d) \in \mathbb{R}^{d+1} \ : \ \|x\|_2 = 1 \right\} \subset \mathbb{R}^{d+1}.$$

The sphere $\mathbb{S}^d$ is a complete metric space with metric $d_{\mathbb{S}^d}$ defined as $d_{\mathbb{S}^d}(a, b) = \arccos \langle a, b \rangle_{\mathbb{R}^{d+1}}$ for $a, b \in \mathbb{S}^d$, where $\langle \cdot, \cdot \rangle_{\mathbb{R}^{d+1}}$ is the standard dot product in $\mathbb{R}^{d+1}$. For $x \in \mathbb{S}^d$, denote $H_x$ be the hyperplane passes through $0 \in \mathbb{R}^{d+1}$ and orthogonal to $x$, i.e. $H_x = \{y \in \mathbb{R}^{d+1} \ : \ \langle x, y \rangle = 0\}$.

We consider the stereographic projection corresponding to $x$, denoted by $\varphi_x$, which is a map from $\mathbb{S}^d \setminus \{x\}$ to $H_x$ defined by: for $y \in \mathbb{S}^d \setminus \{x\}$, $\varphi_x(y)$ is the unique intersection between the line passes through $x, y$ and the hyperplane $H_x$. In concrete, the formula for $\varphi_x$ is as follows

$$\varphi_x : \quad \mathbb{S}^d \setminus \{x\} \longrightarrow H_x$$
$$y \longmapsto \frac{-\langle x, y \rangle}{1 - \langle x, y \rangle} \cdot x + \frac{1}{1 - \langle x, y \rangle} \cdot y. \tag{4}$$

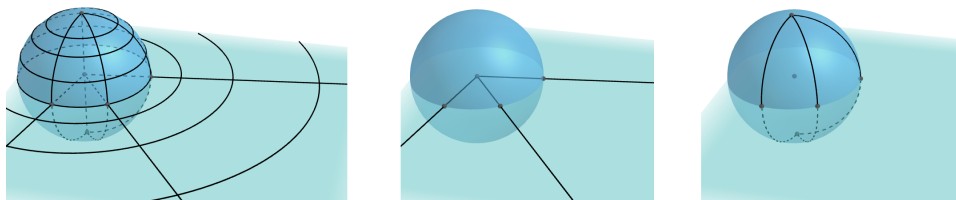

Figure 1: Illustrations of stereographic projection, rays in $\mathbb{R}^3$, and spherical rays in $\mathbb{S}^2$.

It is well-known that $\varphi_x$ is a smooth bijection between $\mathbb{S}^d \setminus \{x\}$ and $H_x$. Moreover, it is convenient to extend $\varphi_x$ to a map that we also denote by $\varphi_x$, from $\mathbb{S}^d$ to $H_x \cup \{\infty\}$, with $\varphi_x(x) = \infty$.

*Remark.* As a topological space, $H_x$ is homeomorphic to $\mathbb{R}^d$, and $H_x \cup \{\infty\}$ is the one-point compactification of $H_x$, which is homeomorphic to $\mathbb{S}^d$. Also, $H_x \cap \mathbb{S}^d$ is homeomorphic to $\mathbb{S}^{d-1}$.

**Definition 3.1** (Spherical rays in $\mathbb{R}^{d+1}$). For $y \in \mathbb{S}^d$, *ray* in $\mathbb{R}^{d+1}$ with direction $y$ is defined as the set $\{t \cdot y : t > 0\} \cup \{\infty\}$. For $x \in \mathbb{S}^d$, and $y \in \mathbb{S}^d \cap H_x$, the *spherical ray* with root $x$ and direction $y$, denoted by $r_y^x$, is defined as the preimage of the ray with direction $y$ through $\varphi_x$, i.e., $r_y^x := \varphi_x^{-1}\big(\{t \cdot y : t > 0\} \cup \{\infty\}\big)$.

An illustration of stereographic projection, rays and spherical rays are presented in Figure 1. In words, a spherical ray with root $x$ and direction $y$ is the great semicircle on surface of the hypersphere passes through $y$ with one endpoint $x$. We have $r_y^x$ is isometric to the closed interval $[0, \pi]$ via $z \mapsto \arccos \langle x, z \rangle$, and we also have a parameterization of $r_y^x$ as $(t, r_y^x)$ for $0 \leqslant t \leqslant \pi$. In particular,

$$\varphi_x^{-1}(0) = -x \mapsto \pi \quad \text{and} \quad \varphi_x^{-1}(\infty) = x \mapsto 0.$$

Let $k$ be a positive integer, $x \in \mathbb{S}^d$ and $y_1, \ldots, y_k \in \mathbb{S}^d \cap H_x$ be $k$ distinct points. We have $k$ distinct spherical rays $r_{y_i}^x$ with root $x$ and direction $y_i$. Consider an equivalence relation $\sim$ on the disjoint union $\bigsqcup_{i=1,\ldots,k} r_{y_i}^x$ as follows: For $(t, r_{y_i}^x) \in r_{y_i}^x$ and $(t', r_{y_j}^x) \in r_{y_j}^x$, we have $(t, r_{y_i}^x) \sim (t', r_{y_j}^x)$ if and only if $(t, r_{y_i}^x) = (t', r_{y_j}^x)$ in $\mathbb{S}^d$. In other words, we identify $k$ points with coordinate 0 on $k$ spherical rays $r_{y_i}^x, 1 \leqslant i \leqslant k$. Denote $\mathcal{T}_{y_1,\ldots,y_k}^x$ as the set of all equivalence classes in $\bigsqcup_{i=1,\ldots,k} r_{y_i}^x$ with respect to the equivalence relation $\sim$, i.e., $\mathcal{T}_{y_1,\ldots,y_k}^x := \big(\bigsqcup_{i=1,\ldots,k} r_{y_i}^x\big)/\sim$.

Recall the notion of disjoint union topology and quotient topology in (Hatcher, 2005). For $i = 1, \ldots, k$, consider the injection

$$
\begin{aligned}
f_i : \quad & r_{y_i}^x \quad \longhookrightarrow \quad \bigsqcup_{i=1,\ldots,k} r_{y_i}^x \\
& (t, r_{y_i}^x) \quad \longmapsto \quad (t, r_{y_i}^x).
\end{aligned}
$$

The disjoint union $\bigsqcup_{i=1,\ldots,k} r_{y_i}^x$ now becomes a topological space with the disjoint union topology, i.e. the finest topology on $\bigsqcup_{i=1,\ldots,k} r_{y_i}^x$ such that the map $f_i$ is continuous for all $i = 1, \ldots, k$. Consider the quotient map by the equivalent relation $\sim$,

$$
\begin{aligned}
\pi : \quad & \bigsqcup_{i=1,\ldots,k} r_{y_i}^x \quad \longrightarrow \quad \mathcal{T}_{y_1,\ldots,y_k}^x = \left( \bigsqcup_{i=1,\ldots,k} r_{y_i}^x \right)/\sim \\
& (t, r_{y_i}^x) \quad \longmapsto \quad [(t, r_{y_i}^x)].
\end{aligned}
$$

$\mathcal{T}_{y_1,\ldots,y_k}^x$ now becomes a topological space with the quotient topology, i.e. the finest topology on $\mathcal{T}_{y_1,\ldots,y_k}^x$ such that the map $\pi$ is continuous. In other words, $\mathcal{T}_{y_1,\ldots,y_k}^x$ is formed by gluing $k$ spherical rays $r_{y_i}^x$ at the points with coordinate 0 on each spherical rays.

**Definition 3.2** (Spherical Trees in $\mathbb{S}^d$). The topological space $\mathcal{T}_{y_1,\ldots,y_k}^x$ is called a *spherical tree* on $\mathbb{S}^d$. We said that $x$ is the root and $y_1, \ldots, y_k$ are the edges of $\mathcal{T}_{y_1,\ldots,y_k}^x$.

A visualization for construction of spherical trees is presented in Figure 2a. The *number of edges* of a spherical tree is usually denoted by $k$. For simplicity, we sometimes omit the root $x$ and edges $y_1, \ldots, y_k$ and simply denote a spherical tree as $\mathcal{T}$. The collection of all spherical trees with $k$ edges

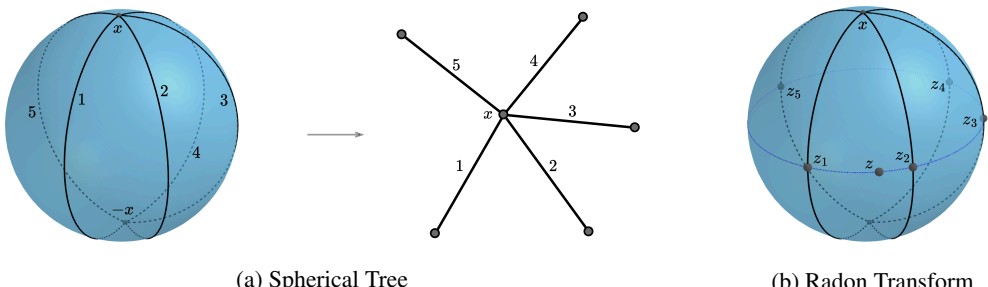

(a) Spherical Tree            (b) Radon Transform

Figure 2: (a) An illustration of 5 spherical rays with the same root $x$, along with the corresponding spherical tree rooted at $x$. Note that, even when endpoints differ from $x$ of these spherical rays are all identical to $-x$ on the sphere, the spherical tree treats these as five distinct points, and only identifies the root $x$. (b) An illustration of Radon Transform on Spherical Trees. Consider a point $z$. The hyperplane passing through $z$ and orthogonal to $x$ cuts edges of the spherical tree at 5 points. The mass at $z$ under operator $\mathcal{R}^\alpha$ is distributed across these 5 intersections, depending on $\alpha$.

on $\mathbb{S}^d$ is denoted by $\mathbb{T}_k^d$. Since $\mathbb{S}^d \cap H_x$ is homeomorphic to the sphere $\mathbb{S}^{d-1}$, we have the one-to-one correspondence between $\mathbb{T}_k^d$ and the product $\mathbb{S}^d \times (\mathbb{S}^d \cap H_x)^k$ as follows:

$$\mathcal{T}_{y_1,\ldots,y_k}^x \quad \xleftrightarrow{\ 1-1\ } \quad x \in \mathbb{S}^d \text{ and } (y_1,\ldots,y_k) \in (\mathbb{S}^d \cap H_x)^k \approx (\mathbb{S}^{d-1})^k. \tag{5}$$

From this observation, we can define a *distribution $\sigma$ on the space of spherical trees* $\mathbb{T}_k^d$ as the joint distribution of distributions on $\mathbb{S}^d$ and $\mathbb{S}^{d-1}$. For the rest of the paper, let $\sigma$ be the joint distribution of $(k+1)$ independent distributions, consists of one uniform distributions on $\mathbb{S}^d$, i.e. $\mathcal{U}(\mathbb{S}^d)$, and $k$ uniform distributions on $\mathbb{S}^{d-1}$, i.e. $\mathcal{U}(\mathbb{S}^{d-1})$. The topological space $\mathcal{T}$ is metrizable by the metric $d_\mathcal{T}$ defined as: For $a = (t, r_{y_i}^x)$ and $b = (t', r_{y_j}^x)$ in $\mathcal{T}$,

$$d_\mathcal{T}(a,b) = \begin{cases} |t - t'|, & \text{if } i = j, \text{ and} \\ t + t', & \text{if } i \neq j. \end{cases} \tag{6}$$

Moreover, this metric is a tree metric on $\mathcal{T}$. We verify this by showing for every pair of two points $a, b$ in $\mathcal{T}$, all paths from $a$ be $b$ in $\mathcal{T}$ are homotopic to each other. Then $d_\mathcal{T}(a,b)$ is the length of the shortest path from $a$ to $b$ in $\mathcal{T}$. Moreover, we can define a measure on $\mathcal{T}$ that induced from the Borel measure on the closed interval $[0, \pi]$. The proof of these properties is similar as the proofs in (Tran et al., 2025d). We summarize our results by a theorem.

**Theorem 3.3** (Spherical trees are metric spaces with tree metric). *$\mathcal{T}$ is a metric space with tree metric $d_\mathcal{T}$. The topology on $\mathcal{T}$ induced by $d_\mathcal{T}$ is identical to the topology of $\mathcal{T}$.*

With this design, in the next section, we will define Lebesgue integrable functions on spherical trees.

## 4   SPHERICAL RADON TRANSFORM ON SPHERICAL TREES

In this section, we introduce the spherical Radon Transform on Spherical Trees, and discuss the injectivity of our spherical Radon transform variant.

### 4.1   A SPHERICAL RADON TRANSFORM VARIANT

We introduce the notions of the space of Lebesgue integrable functions on spherical trees. First, denote $L^1(\mathbb{S}^d)$ as the space of Lebesgue integrable functions on $\mathbb{S}^d$ with norm $\|\cdot\|_1$:

$$L^1(\mathbb{S}^d) = \left\{ f \colon \mathbb{S}^d \to \mathbb{R} \ : \ \|f\|_1 = \int_{\mathbb{S}^d} |f(x)| \, dx < \infty \right\}. \tag{7}$$

Two functions $f_1, f_2 \in L^1(\mathbb{S}^d)$ are considered to be identical if $f_1(x) = f_2(x)$ for almost everywhere on $\mathbb{S}^d$. Consider a spherical tree $\mathcal{T}$ with root $x$ and $k$ edges $y_1,\ldots,y_k$, a *Lebesgue integrable function on $\mathcal{T}$* is a function $f \colon \mathcal{T} \to \mathbb{R}$ such that $\|f\|_\mathcal{T} := \sum_{i=1}^{k} \int_0^\pi |f(t, r_{y_i}^x)| \, dt < \infty$.

The *space of Lebesgue integrable functions on $\mathcal{T}$* is denoted by $L^1(\mathcal{T})$. Two functions $f_1, f_2 \in L^1(\mathcal{T})$ are considered to be identical if $f_1(x) = f_2(x)$ for almost everywhere on $\mathcal{T}$. The space $L^1(\mathcal{L})$ with norm $\|\cdot\|_\mathcal{T}$ is a Banach space.

Let $\Delta_{k-1} := \left\{ (a_i)_{i=1}^k : 0 \leqslant a_i \leqslant 1 \text{ and } \sum_{i=1}^k a_i = 1 \right\} \subset \mathbb{R}^k$ be the $(k-1)$-dimensional standard simplex. Denote $\mathcal{C}\left(\mathbb{S}^d \times \mathbb{T}_k^d, \Delta_{k-1}\right)$ as the space of continuous maps from $\mathbb{S}^d \times \mathbb{T}_k^d$ to $\Delta_{k-1}$, and called a map in $\mathcal{C}\left(\mathbb{S}^d \times \mathbb{T}_k^d, \Delta_{k-1}\right)$ by a *splitting map*. Let $\mathcal{T}$ be a spherical tree with root $x$ and $k$ edges $y_1, \ldots, y_k$, $\alpha$ be a splitting map in $\mathcal{C}\left(\mathbb{S}^d \times \mathbb{T}_k^d, \Delta_{k-1}\right)$, we define an operator associated to $\alpha$ that transforms a Lebesgue integrable functions on $\mathbb{S}^d$ to a Lebesgue integrable functions on $\mathcal{T}$. For $f \in L^1(\mathbb{S}^d)$, define

$$\mathcal{R}_\mathcal{T}^\alpha f : \mathcal{T} \longrightarrow \mathbb{R} \tag{8}$$

$$(t, r_{y_i}^x) \longmapsto \int_{\mathbb{S}^d} f(y) \cdot \alpha(y, \mathcal{T})_i \cdot \delta(t - \arccos \langle x, y \rangle) \, dy, \tag{9}$$

where $\delta$ is the Dirac delta function. We have $\mathcal{R}_\mathcal{T}^\alpha f \in L^1(\mathcal{T})$ for $f \in L^1(\mathbb{S}^d)$, and moreover, $\|\mathcal{R}_\mathcal{T}^\alpha f\|_\mathcal{T} \leqslant \|f\|_1$. The operator $\mathcal{R}_\mathcal{T}^\alpha : L^1(\mathbb{S}^d) \to L^1(\mathcal{T})$ is a well-defined linear operator. The proof of these properties can be found in Appendix A.1. An illustration of $\mathcal{R}_\mathcal{T}^\alpha$ is presented in Figure 2b. We next present a novel spherical Radon Transform variant on spherical trees.

**Definition 4.1** (Spherical Radon Transform on Spherical Trees). For $\alpha \in \mathcal{C}\left(\mathbb{S}^d \times \mathbb{T}_k^d, \Delta_{k-1}\right)$, the operator $\mathcal{R}^\alpha$ that is defined as follows:

$$\mathcal{R}^\alpha : L^1(\mathbb{S}^d) \longrightarrow \prod_{\mathcal{T} \in \mathbb{T}_k^d} L^1(\mathcal{T})$$

$$f \longmapsto (\mathcal{R}_\mathcal{T}^\alpha f)_{\mathcal{T} \in \mathbb{T}_k^d}.$$

is called the *Spherical Radon Transform on Spherical Trees*.

## 4.2 INJECTIVITY OF RADON TRANSFORM ON SPHERICAL TREES

We discuss on the injectivity of our spherical Radon Transform variant. Consider the Euclidean norm on $\mathbb{R}^d$, i.e. $\|\cdot\|_2$.

**Orthogonal group $\mathrm{O}(d)$ and its actions.** The orthogonal group $\mathrm{O}(d)$ is the group of linear transformations of $\mathbb{R}^d$ that preserves the Euclidean norm $\|\cdot\|_2$,

$$\mathrm{O}(d) = \left\{ \text{linear transformation } f : \mathbb{R}^d \to \mathbb{R}^d : \|x\|_2 = \|f(x)\|_2 \text{ for all } x \in \mathbb{R}^d \right\}. \tag{10}$$

It is well-known that $\mathrm{O}(d)$ is isomorphic to the group of orthogonal matrices under multiplication,

$$\mathrm{O}(d) = \left\{ Q \in M_{d \times d}(\mathbb{R}) : Q \cdot Q^\top = Q^\top \cdot Q = I_d \right\}. \tag{11}$$

The canonical group action of $\mathrm{O}(d)$ on $\mathbb{R}^d$ is defined by: For $g = Q \in \mathrm{O}(d)$ and $y \in \mathbb{R}^d$, we have $y \mapsto gy = Q \cdot y$. By the norm preserving, the action of $\mathrm{O}(d+1)$ on $\mathbb{R}^{d+1}$ canonically induces an action of $\mathrm{O}(d+1)$ on the sphere $\mathbb{S}^d$. Moreover, the action of $\mathrm{O}(d)$ on $\mathbb{R}^d$ preserves the standard dot product, so the action of $\mathrm{O}(d+1)$ on $\mathbb{S}^d$ preserves the metric $d_{\mathbb{S}^d}$.

**Group actions of $\mathrm{O}(d+1)$ on space of spherical trees $\mathbb{T}_k^d$.** Under $g \in \mathrm{O}(d+1)$, the spherical ray $r_y^x$ transforms to $r_{gy}^{gx}$. It implies that the action of $\mathrm{O}(d+1)$ on $\mathbb{S}^d$ canonically induces an action of $\mathrm{O}(d+1)$ on $\mathbb{T}_k^d$ as

$$\mathcal{T} = \mathcal{T}_{y_1, \ldots, y_k}^x \longmapsto g\mathcal{T} := \mathcal{T}_{gy_1, \ldots, gy_k}^{gx}. \tag{12}$$

Moreover, each $g \in \mathrm{O}(d+1)$ presents a morphism $\mathcal{T} \to g\mathcal{T}$ that is isometric.

$\mathrm{O}(d+1)$**-invariant splitting maps.** Given a map $f : X \to Y$ and a group $G$ acts on $X$. The map $f$ is called $G$-invariant if $f(gx) = f(x)$ for all $g \in G$ and $x \in X$. We have the definition of $\mathrm{O}(d+1)$-invariance in splitting maps.

**Definition 4.2.** A splitting map $\alpha$ in $\mathcal{C}(\mathbb{S}^d \times \mathbb{T}_k^d, \Delta_{k-1})$ is said to be $\mathrm{O}(d+1)$-invariant, if we have

$$\alpha(gy, g\mathcal{T}) = \alpha(y, \mathcal{T}) \tag{13}$$

for all $(y, \mathcal{T}) \in \mathbb{S}^d \times \mathbb{T}_k^d$ and $g \in \mathrm{O}(d+1)$.

With an $\mathrm{O}(d+1)$-invariant splitting maps, our spherical Radon Transform variant is injective.

**Theorem 4.3.** $\mathcal{R}^\alpha$ *is injective for an* $\mathrm{O}(d+1)-$*invariant splitting map* $\alpha$.

The proof of Theorem 4.3 is presented in Appendix A.3. Finally, we present a candidate for $\mathrm{O}(d+1)$-invariant splitting maps. Define the map $\beta\colon \mathbb{S}^d \times \mathbb{T}_k^d \to \mathbb{R}^k$ as follows:

$$\beta(y, \mathcal{T}_{y_1,\ldots,y_k}^x)_i = \begin{cases} 0, & \text{if } y = x \text{ or } y = -x, \\ \arccos\left(\dfrac{\langle y, y_i \rangle}{\sqrt{1 - \langle x, y \rangle^2}}\right) \cdot \sqrt{1 - \langle x, y \rangle^2}, & \text{if } y \neq \pm x. \end{cases} \tag{14}$$

*Remark.* The construction of $\beta$ will be explained in Appendix A.2.

The map $\beta$ is continuous and $\mathrm{O}(d+1)$-invariant. The derivation of $\beta$ and the proof for these properties are presented in Appendix A.2. We choose $\alpha\colon \mathbb{S}^d \times \mathbb{T}_k^d \to \Delta_{k-1}$ as follows:

$$\alpha(y, \mathcal{T}) = \mathrm{softmax}\Big(\{\zeta \cdot \beta(y, \mathcal{T})_i\}_{i=1,\ldots,k}\Big) \tag{15}$$

Here, $\zeta \in \mathbb{R}$ is treated as a tuning parameter. The intuition behind this choice of $\alpha$ is that it reflects the proximity of points to the rays of the spherical trees. As $|\zeta|$ increases, the resulting value of $\alpha$ tends to become more sparse, emphasizing the importance of each ray relative to a specific point.

## 5 SPHERICAL TREE-SLICED WASSERSTEIN DISTANCE

In this section, we propose our novel Spherical Tree-Sliced Wasserstein Distance (STSW). We also derive a closed-form approximation of STSW that allows an efficient implementation.

### 5.1 SPHERICAL TREE-SLICED WASSERSTEIN DISTANCE

Given two probability distributions $\mu, \nu$ in $\mathcal{P}(\mathbb{S}^d)$, a tree $\mathcal{T} \in \mathbb{T}_k^d$ and an $\mathrm{O}(d+1)$-invariant splitting map $\alpha \in \mathcal{C}(\mathbb{S}^d \times \mathbb{T}_k^d, \Delta_{k-1})$. By the Radon Transform $\mathcal{R}_{\mathcal{T}}^\alpha$ in Definition 4.1, $\mu$ and $\nu$ tranform to two probability distributions $\mathcal{R}_{\mathcal{L}}^\alpha \mu$ and $\mathcal{R}_{\mathcal{L}}^\alpha \nu$ in $\mathcal{P}(\mathcal{T})$. $\mathcal{T}$ is a metric space with tree metric $d_{\mathcal{T}}$ (Tran et al., 2025d), so we can compute Wasserstein distance $\mathrm{W}_{d_{\mathcal{T}},1}(\mathcal{R}_{\mathcal{T}}^\alpha \mu, \mathcal{R}_{\mathcal{T}}^\alpha \nu)$ between $\mathcal{R}_{\mathcal{T}}^\alpha \mu$ and $\mathcal{R}_{\mathcal{T}}^\alpha \nu$ by Equation (3).

**Definition 5.1** (Spherical Tree-Sliced Wasserstein Distance)**.** The *Spherical Tree-Sliced Wasserstein Distance* between $\mu, \nu$ in $\mathcal{P}(\mathbb{S}^d)$ is defined by:

$$\mathrm{STSW}(\mu, \nu) \coloneqq \int_{\mathbb{T}_k^d} \mathrm{W}_{d_{\mathcal{T}},1}(\mathcal{R}_{\mathcal{T}}^\alpha \mu, \mathcal{R}_{\mathcal{T}}^\alpha \nu)\, d\sigma(\mathcal{T}). \tag{16}$$

*Remark.* Note that, the definition of STSW depends on the space $\mathbb{T}_k^d$, the distribution $\sigma$ on $\mathbb{T}_k^d$, and the splitting map $\alpha$ as in Equation (15). We omit them to simplify the notation.

The STSW distance is, indeed, a metric on $\mathcal{P}(\mathbb{S}^d)$.

**Theorem 5.2.** STSW *is a metric on* $\mathcal{P}(\mathbb{S}^d)$. *Moreover,* STSW *is invariant under orthogonal transformations: For* $g \in \mathrm{O}(d+1)$*, we have*

$$\mathrm{STSW}(\mu, \nu) = \mathrm{STSW}(g\sharp\mu, g\sharp\nu), \tag{17}$$

*where* $g\sharp\mu, g\sharp\nu$ *as the push-forward of* $\mu, \nu$ *via orthogonal transformation* $g\colon \mathbb{S}^d \to \mathbb{S}^d$*, respectively.*

The proofs of Theorem 5.2 is presented in Appendix A.4.

### 5.2 COMPUTATION OF STSW

To approximate the intractable integral in Equation (16), we use the Monte Carlo method as $\widehat{\mathrm{STSW}}(\mu, \nu) = (1/L) \cdot \Sigma_{l=1}^L \mathrm{W}_{d_{\mathcal{T}_l},1}(\mathcal{R}_{\mathcal{T}_l}^\alpha \mu, \mathcal{R}_{\mathcal{T}_l}^\alpha \nu)$, where $\mathcal{T}_1, \ldots, \mathcal{T}_L$ are drawn independently from the distribution $\sigma$ on $\mathbb{T}$, and are referred to as projecting tree systems. We present the way to sample $\mathcal{T}_i$ and compute $\mathrm{W}_{d_{\mathcal{T}_l},1}(\mathcal{R}_{\mathcal{T}_l}^\alpha \mu, \mathcal{R}_{\mathcal{T}_l}^\alpha \nu)$.

**Sampling spherical trees.** Recall that $\sigma$ is the joint distribution of $k+1$ independent distributions, consists of one uniform distributions on $\mathbb{S}^d$, and $k$ uniform distributions on $\mathbb{S}^{d-1}$. This comes from the one-to-one correspondence between $\mathbb{T}_k^d$ and $\mathbb{S}^d \times (\mathbb{S}^{d-1})^k$ as in Equation (5). In applications, to perform a sampling process for $\mathcal{T} = \mathcal{T}_{y_1,\ldots,y_k}^x \in \mathbb{T}_k^d$ from $\sigma$, we sample by two steps as follows:

1. Sample $k+1$ points $x, y_1, \ldots, y_k$ in $\mathbb{R}^{d+1}$. Normalize them to get $x, y_1, \ldots, y_k$ lie on $\mathbb{S}^d$.

2. For each $i$, take the intersection of the line passes through $x, y_i$ with $H_x$, i.e. $\varphi_x$, then normalize $\Phi_x$ to get new $y_i$ lies on $H_x \cap \mathbb{S}^d$.

This results in a sampling process based on distribution $\sigma$.

**Computing** $W_{d_\mathcal{T},1}(\mathcal{R}_\mathcal{T}^\alpha \mu, \mathcal{R}_\mathcal{T}^\alpha \nu)$**.** In applications, given discrete distributions $\mu$ and $\nu$ as $\mu(x) = \sum_{j=1}^n u_j \cdot \delta(x - a_j)$ and $\nu(x) = \sum_{j=1}^n v_j \cdot \delta(x - a_j)$. We can present $\mu$ and $\nu$ with the same supports by combining their supports and allow some $u_j$ or $v_j$ to be 0. For spherical tree $\mathcal{T} = \mathcal{T}_{y_1,\ldots,y_k}^x$, we want to compute $W_{d_\mathcal{T},1}(\mathcal{R}_\mathcal{T}^\alpha \mu, \mathcal{R}_\mathcal{T}^\alpha \nu)$. For $1 \leqslant j \leqslant n$, let $c_j = d_{\mathbb{S}^d}(x, a_j)$. Also let $c_0 = 0$. By re-indexing, we assume that $0 = c_0 \leqslant c_1 \leqslant \ldots \leqslant c_n$. By Radon Transform in Definition 4.1, $\mu, \nu$ transform to $\mathcal{R}_\mathcal{T}^\alpha \mu, \mathcal{R}_\mathcal{T}^\alpha \nu$ supported on $\{(c_j, r_{y_i}^x)\}_{1 \leqslant i \leqslant k, 1 \leqslant j \leqslant n}$ of $\mathcal{T}$, with

$$\mathcal{R}_\mathcal{T}^\alpha \mu(c_j, r_{y_i}^x) = \alpha(a_j, \mathcal{T})_i \cdot u_j \quad \text{and} \quad \mathcal{R}_\mathcal{T}^\alpha \nu(c_j, r_{y_i}^x) = \alpha(a_j, \mathcal{T})_i \cdot v_j \tag{18}$$

By Equation (3), $W_{d_\mathcal{T},1}(\mathcal{R}_\mathcal{T}^\alpha \mu, \mathcal{R}_\mathcal{T}^\alpha \nu)$ has a closed-form approximation as follows

$$W_{d_\mathcal{T},1}(\mathcal{R}_\mathcal{T}^\alpha \mu, \mathcal{R}_\mathcal{T}^\alpha \nu) = \sum_{j=1}^n (c_j - c_{j-1}) \cdot \left( \sum_{i=1}^k \left| \sum_{p=j}^n \alpha(a_p, \mathcal{T})_i \cdot (u_p - v_p) \right| \right). \tag{19}$$

The detailed derivation of Equation (19) is presented in Appendix A.5. The closed-form expression in Equation (19) leads to a highly parallelizable implementation of STSW distance.[1]

## 6 EXPERIMENTAL RESULTS

In this section, we present the results of our four main tasks: Gradient Flow, Self-Supervised Learning, Earth Density Estimation, and Sliced-Wasserstein Auto-Encoder. We provide a detailed evaluation for each task, including quantitative metrics, visualizations, and a comparison with relevant baseline methods. Experimental details can be found in Appendix B.

### 6.1 GRADIENT FLOW

Our first experiment focuses on learning a target distribution $\nu$ from a source distribution $\mu$ by minimizing STSW($\nu, \mu$). We solve this optimization using projected gradient descent, as discussed in Bonet et al. (2022). We compare the performance of our method against baselines: SSW (Bonet et al., 2022), and S3W variants (Tran et al., 2024b).

Following Tran et al. (2024b), we use a mixture of 12 von Mises-Fisher distributions (vMFs) as our target $\nu$. The training is conducted over 500 epochs with a full batch, and each experiment is repeated 10 times. We adopt the evaluation metrics from Tran et al. (2024b), which include log 2-Wasserstein distance, negative log-likelihood (NLL), and training time. As shown in Table 1, STSW outperforms the baselines in all metrics and achieves faster convergence, as illustrated in Figure 10.

### 6.2 SELF-SUPERVISED LEARNING (SSL)

Normalizing feature vectors to the hypersphere has been shown to improve the quality of learned representations and prevent feature collapse (Chen et al., 2020; Wang & Isola, 2020). In previous work, Wang & Isola (2020) identified two properties of contrastive learning: alignment (bringing positive pairs closer) and uniformity (distributing features evenly on the hypersphere). Adopting

---

[1]See Algorithm 1 in Appendix §B.1 for pseudo-code of STSW distance.

Table 1: Learning target distribution 12 vFMs. We use $N_R = 30$ rotations for ARI-S3W and an additional learning rate LR = 0.05 for SSW.

| Method | $\log W_2 \downarrow$ | NLL $\downarrow$ | Runtime(s) |
|---|---|---|---|
| SSW (LR=0.01) | -3.21 ± 0.16 | -4980.01 ± 1.89 | 55.20 ± 0.15 |
| SSW (LR=0.05) | -3.36 ± 0.12 | -4976.58 ± 2.23 | 55.31 ± 0.33 |
| S3W | -2.37 ± 0.21 | -4749.67 ± 84.34 | 1.93 ± 0.06 |
| RI-S3W (1) | -3.12 ± 0.18 | -4964.50 ± 27.98 | 2.03 ± 0.12 |
| RI-S3W (5) | -3.47 ± 0.06 | -4984.80 ± 7.32 | 5.68 ± 0.51 |
| ARI-S3W (30) | -4.39 ± 0.19 | -5020.37 ± 6.35 | 20.25 ± 0.15 |
| STSW | **-4.69 ± 0.01** | **-5041.13 ± 0.84** | **1.89 ± 0.05** |

Table 2: CIFAR-10 linear evaluation accuracy for encoded (E) features and projected (P) features on $\mathbb{S}^9$, along with pretraining time per epoch. ARI-S3W and RI-S3W use 5 rotations.

| Method | Acc. E(%) ↑ | Acc. P(%) ↑ | Time (s/ep.) |
|---|---|---|---|
| Hypersphere | 79.81 | 74.64 | 10.18 |
| SimCLR | 79.97 | 72.80 | **9.34** |
| SW | 74.39 | 67.80 | 9.65 |
| SSW | 70.23 | 64.33 | 10.59 |
| S3W | 78.59 | 73.83 | 10.14 |
| RI-S3W (5) | 79.93 | 73.95 | 10.22 |
| ARI-S3W (5) | 80.08 | 75.12 | 10.19 |
| STSW | **80.53** | **76.78** | 9.54 |

the approach in Bonet et al. (2022), we propose replacing the Gaussian kernel uniformity loss with STSW, resulting in the following contrastive objective:

$$\mathcal{L} = \underbrace{\frac{1}{n}\sum_{i=1}^{n}\left\|z_i^A - z_i^B\right\|_2^2}_{\text{Alignment loss}} + \frac{\lambda}{2}\underbrace{\left(\text{STSW}(z^A,\nu) + \text{STSW}(z^B,\nu)\right)}_{\text{Uniformity loss}}, \qquad (20)$$

where $\nu = \mathcal{U}(\mathbb{S}^d)$, $\lambda > 0$ is regularization factor, $z^A, z^B \in \mathbb{R}^{n\times(d+1)}$ are embeddings of two image augmentations mapped onto $\mathbb{S}^d$. Similar to Bonet et al. (2022) and Tran et al. (2024b), we train ResNet18 (He et al., 2016) based encoder on the CIFAR-10 (Krizhevsky et al., 2009) w.r.t $\mathcal{L}$. After this, we train a linear classifier on the features extracted from the pre-trained encoder.

Table 2 demonstrates the improvement of STSW in comparison to baselines: Hypersphere (Wang & Isola, 2020), SimCLR (Chen et al., 2020), SW, SSW, and S3W variants. We also conduct experiments with $d = 2$ to visualize learned representations. Figure 12 illustrates that STSW can effectively distribute encoded features around the sphere while keeping similar ones close together.

### 6.3 EARTH DENSITY ESTIMATION

We now demonstrate the application of STSW in density estimation on $\mathbb{S}^2$. Data used in this task is collected by (Mathieu & Nickel, 2020) which consists of Fire (Brakenridge, 2017), Earthquake (EOSDIS, 2020) and Flood (EOSDIS, 2020). As in (Bonet et al., 2022), we employ an exponential map normalizing flow model (Rezende et al., 2020) which are invertible transformations $T$ and aim to minimize $\min_T \text{STSW}(T_{\#}\mu, p_Z)$, where $\mu$ is the empirical distribution, and $p_Z$ is a prior distribution on $\mathbb{S}^2$ which we use uniform distribution. The density for any point $x \in \mathbb{S}^2$ is then estimated by $f_\mu(x) = p_Z(T(x))|\det J_T(x)|$ where $J_T(x)$ is the Jacobian of $T$ at $x$.

Our baselines are exponential map normalizing flows with SW, SSW, and S3W variants, and stereographic projection-based (Dinh et al., 2016) normalizing flows. As seen in Table 3, STSW even with fewer epochs and shorter training time (10K epochs over 2h10m for STSW versus 20K epochs

Table 3: Negative log-likelihood on test data, averaged over 5 runs with different data split.

| Method | Quake ↓ | Flood ↓ | Fire ↓ |
|---|---|---|---|
| Stereo | $1.94 \pm 0.21$ | $1.92 \pm 0.04$ | $1.31 \pm 0.12$ |
| SW | $0.99 \pm 0.05$ | $1.47 \pm 0.03$ | $0.55 \pm 0.21$ |
| SSW | $0.84 \pm 0.06$ | $1.26 \pm 0.04$ | $0.23 \pm 0.20$ |
| S3W | $0.89 \pm 0.08$ | $1.35 \pm 0.04$ | $0.34 \pm 0.05$ |
| RI-S3W (1) | $0.80 \pm 0.07$ | $1.25 \pm 0.03$ | $0.14 \pm 0.06$ |
| ARI-S3W (50) | $0.77 \pm 0.06$ | $1.24 \pm 0.03$ | $0.08 \pm 0.05$ |
| STSW | $\mathbf{0.68 \pm 0.04}$ | $\mathbf{1.23 \pm 0.03}$ | $\mathbf{-0.07 \pm 0.05}$ |

Table 4: SWAE results evaluated on latent regularization of CIFAR-10 test data.

| Method | $\log W_2$ ↓ | NLL ↓ | BCE ↓ | Time (s/ep.) |
|---|---|---|---|---|
| SW | -3.2943 | -0.0014 | 0.6314 | **3.4060** |
| SSW | -2.2234 | 0.0005 | **0.6309** | 8.2386 |
| S3W | -3.3421 | 0.0013 | 0.6329 | 4.5138 |
| RI-S3W (5) | -3.1950 | -0.0039 | 0.6354 | 4.9682 |
| ARI-S3W (5) | -3.3935 | 0.0012 | 0.6330 | 4.7347 |
| STSW | **-3.4191** | **-0.0051** | 0.6341 | 3.5460 |

over 4h30m for ARI-S3W, both on Fire dataset) still outperforms or is competitive with SSW and S3W variants.

### 6.4 SLICED-WASSERSTEIN AUTO-ENCODER (SWAE)

We apply STSW to generative modeling using the Sliced-Wasserstein Auto-Encoder (SWAE) (Kolouri et al., 2018) framework, which regularizes the latent space distribution to match a prior distribution $q$. Let $\varphi : \mathcal{X} \to \mathbb{S}^d$ and $\psi : \mathbb{S}^d \to \mathcal{X}$ be the parametric encoder and decoder. The objective of the SWAE is $\min_{\varphi, \psi} \mathbb{E}_{x \sim p} \left[ c(x, \psi(\varphi(x))) \right] + \lambda \cdot \text{STSW} \left( \varphi_{\#} p, q \right)$, where $\lambda$ controls regularization, $p$ is data distribution. We use SW, SSW (Bonet et al., 2022) and S3W variants (Tran et al., 2024b) as baselines, Binary Cross Entropy (BCE) for reconstruction loss and a mixture of 10 vMFs as the prior, similar to Tran et al. (2024b). We provide results in Table 4. We note that STSW has the best results in log 2-Wasserstein and NLL with a competitive training time, though its BCE slightly underperforms the others.

## 7 CONCLUSION

This paper introduces the Spherical Tree-Sliced Wasserstein (STSW) distance, a novel approach leveraging a new integration domain called spherical trees. In contrast to the traditional one-dimensional lines or great semicircles often used in the spherical Sliced Wasserstein variant, STSW ultilizes spherical trees to better capture the topology of spherical data and provides closed-form solutions for optimal transport problems on spherical trees, leading to expected improvements in both performance and efficiency. We rigorously develop the theoretical basis for our approach by introducing spherical Radon Transform on Spherical Tree then verifying the core properties of the transform such as its injectivity. We thoroughly develop the theoretical foundation for this method by introducing the spherical Radon Transform on Spherical Trees and validating its key properties, such as injectivity. STSW is derived from the Radon Transform framework, and through careful construction of the splitting maps, we obtain a closed-form approximation for the distance. Through empirical tasks on spherical data, we demonstrate that STSW significantly outperforms recent spherical Wasserstein variants. Future research could explore spherical trees further, such as developing sampling processes for spherical trees or adapting Generalized Radon Transforms to enhance STSW.

ACKNOWLEDGMENTS

This research / project is supported by the National Research Foundation Singapore under the AI Singapore Programme (AISG Award No: AISG2-TC-2023-012-SGIL). This research / project is supported by the Ministry of Education, Singapore, under the Academic Research Fund Tier 1 (FY2023) (A-8002040-00-00, A-8002039-00-00). This research / project is also supported by the NUS Presidential Young Professorship Award (A-0009807-01-00) and the NUS Artificial Intelligence Institute–Seed Funding (A-8003062-00-00).

We thank the area chairs, anonymous reviewers for their comments. TL acknowledges the support of JSPS KAKENHI Grant number 23K11243, and Mitsui Knowledge Industry Co., Ltd. grant.

**Ethics Statement.** Given the nature of the work, we do not foresee any negative societal and ethical impacts of our work.

**Reproducibility Statement.** Source codes for our experiments are provided in the supplementary materials of the paper. The details of our experimental settings and computational infrastructure are given in Section 6 and the Appendix. All datasets that we used in the paper are published, and they are easy to access in the Internet.

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

NOTATION

| | |
|---|---|
| $\mathbb{R}^d$ | $d$-dimensional Euclidean space |
| $\|\cdot\|_2$ | Euclidean norm |
| $\langle\cdot,\cdot\rangle$ | standard dot product |
| $\mathbb{S}^d$ | $d$-dimensional hypersphere |
| $\theta$ | unit vector |
| $\sqcup$ | disjoint union |
| arccos | inverse of cosine function |
| $L^1(X)$ | space of Lebesgue integrable functions on $X$ |
| $\mathcal{P}(X)$ | space of probability distributions on $X$ |
| $\mu, \nu$ | measures |
| $\delta(\cdot)$ | 1-dimensional Dirac delta function |
| $\mathcal{U}(\mathbb{S}^d)$ | uniform distribution on $\mathbb{S}^d$ |
| $\sharp$ | pushforward (measure) |
| $\mathcal{C}(X, Y)$ | space of continuous maps from $X$ to $Y$ |
| $d(\cdot, \cdot)$ | metric in metric space |
| $\mathrm{O}(d)$ | orthogonal group of order $d$ |
| $g$ | element of group |
| $\mathrm{W}_p$ | $p$-Wasserstein distance |
| $\mathrm{SW}_p$ | Sliced $p$-Wasserstein distance |
| $\Gamma$ | (rooted) subtree |
| $e$ | edge in graph |
| $w_e$ | weight of edge in graph |
| $\varphi_x$ | stereographic projection at $x$ |
| $H_x$ | hyperplane passes through $x$ and orthogonal to $x$ |
| $r_y^x$ | spherical ray |
| $\mathcal{T}, \mathcal{T}_{y_1,\ldots,y_k}^x$ | spherical tree |
| $\mathbb{T}_k^d$ | space of spherical trees of $k$ edges on $\mathbb{S}^d$ |
| $\sigma$ | distribution on space of tree systems |
| $L$ | number of spherical tree |
| $k$ | number of edges in spherical tree |
| $\mathcal{R}$ | original Radon Transform |
| $\mathcal{R}^\alpha$ | spherical Radon Transform on Spherical Trees |
| $\Delta_{k-1}$ | $(k-1)$-dimensional standard simplex |
| $\alpha$ | splitting map |
| $\zeta$ | tuning parameter in splitting maps |

# Supplement to
## "Spherical Tree-Sliced Wasserstein Distance"

**Table of Contents**

## A  THEORETICAL PROOFS

### A.1  PROPERTIES OF $\mathcal{R}_{\mathcal{T}}^{\alpha} f$

*Proof.* Let $f \in L^1(\mathbb{S}^d)$. We show that $\|\mathcal{R}_{\mathcal{T}}^{\alpha} f\|_{\mathcal{T}} \leqslant \|f\|_1$. Note that, $\arccos \langle x, y \rangle \in [0, \pi]$, so we have

$$
\begin{aligned}
\|\mathcal{R}_{\mathcal{T}}^{\alpha} f\|_{\mathcal{T}} &= \sum_{i=1}^{k} \int_0^{\pi} \left| \mathcal{R}_{\mathcal{T}}^{\alpha} f(t, r_{y_i}^x) \right| \, dt \\
&= \sum_{i=1}^{k} \int_0^{\pi} \left| \int_{\mathbb{S}^d} f(y) \cdot \alpha(y, \mathcal{T})_i \cdot \delta(t - \arccos \langle x, y \rangle) \, dy \right| \, dt \\
&\leqslant \sum_{i=1}^{k} \int_0^{\pi} \left( \int_{\mathbb{S}^d} |f(y)| \cdot \alpha(y, \mathcal{T})_i \cdot \delta(t - \arccos \langle x, y \rangle) \, dy \right) \, dt \\
&= \sum_{i=1}^{k} \int_{\mathbb{S}^d} \left( \int_0^{\pi} |f(y)| \cdot \alpha(y, \mathcal{T})_i \cdot \delta(t - \arccos \langle x, y \rangle) \, dt \right) \, dy \\
&= \sum_{i=1}^{k} \int_{\mathbb{S}^d} |f(y)| \cdot \alpha(y, \mathcal{T})_i \cdot \left( \int_0^{\pi} \delta(t - \arccos \langle x, y \rangle) \, dt \right) \, dy \\
&= \sum_{i=1}^{k} \int_{\mathbb{S}^d} |f(y)| \cdot \alpha(y, \mathcal{T})_i \, dy \\
&= \int_{\mathbb{S}^d} |f(y)| \cdot \left( \sum_{i=1}^{k} \alpha(y, \mathcal{T})_i \right) \, dy \\
&= \int_{\mathbb{S}^d} |f(y)| \, dy
\end{aligned}
$$

$$= \|f\|_1 < \infty.$$

It implies that $\mathcal{R}_{\mathcal{T}}^{\alpha} f \in L^1(\mathcal{T})$, which means the operator $\mathcal{R}_{\mathcal{T}}^{\alpha} \colon L^1(\mathbb{S}^d) \to L^1(\mathcal{T})$ is well-defined. Clearly, $\mathcal{R}_{\mathcal{T}}^{\alpha}$ is a linear operator. $\qquad\square$

## A.2 DERIVATION AND PROPERTIES OF SPLITTING MAPS

**Invariance and Equivariance Properties in Machine Learning.** Equivariant networks (Cohen & Welling, 2016) improve generalization and boost sample efficiency by incorporating task symmetries directly into their architecture. These networks have been notably successful in several fields, including trajectory prediction (Walters et al., 2020), robotics (Simeonov et al., 2022), graph-based models (Satorras et al., 2021; Tran et al., 2024a), and functional networks (Tran et al., 2025a; 2024c; Vo et al., 2024), among others. The use of equivariance has been demonstrated to enhance performance, increase data efficiency, and strengthen robustness against out-of-domain generalization.

We recall the construction for a splitting map $\alpha$ presented in Subsection 4.2. We have a map $\beta \colon \mathbb{S}^d \times \mathbb{T}_k^d \to \mathbb{R}^k$ defined as follows:

$$\beta(y, \mathcal{T}_{y_1, \dots, y_k}^x)_i = \begin{cases} 0, & \text{if } y = x \text{ or } y = -x, \text{ and} \\ \arccos\left( \dfrac{\langle y, y_i \rangle}{\sqrt{1 - \langle x, y \rangle^2}} \right) \cdot \sqrt{1 - \langle x, y \rangle^2}, & \text{if } y \neq x, -x. \end{cases} \tag{21}$$

Then $\alpha \colon \mathbb{S}^d \times \mathbb{T}_k^d \to \Delta_{k-1}$ is defined as follows:

$$\alpha(y, \mathcal{T}) = \text{softmax}\left( \{\delta \cdot \beta(y, \mathcal{T})_i\}_{i=1, \dots, k} \right) \tag{22}$$

We will show that

1. Where does $\beta$ come from?
2. $\alpha$ is continuous.
3. $\alpha$ is $\mathrm{O}(d+1)$-invariant.

*Proof.* We prove each part.

**1.** For $(y, \mathcal{T}_{y_1, \dots, y_k}^x) \in \mathbb{S}^d \times \mathbb{T}_k^d$, let $N_y$ be the hyperplane passes through $Y$ and orthogonal to $x$. Then $N_y$ intersects the spherical ray $r_{y_i}^x$ at a single point $a$, and intersects the vector $x$ at a single point $b$. The $\beta(y, \mathcal{T})_i$ is the length of the small arc from $y$ to $a$ on the circle centered at $b$ passes through $y$ and $a$. Indeed, if $y = x$ and $y = -x$, this length is equal to 0, the same as the definition of $\beta$. If $y \neq x, -x$, let $c$ is the intersection of the line passes through $x, y$, and the hyperplane $H_x$; moreover, let $d$ be the unique intersection of the segment with endpoints $0, c$, and the hyperplane $H_x$. In details, we have

$$c = \varphi_x(y) \text{ and } d = \frac{c}{\|c\|_2}. \tag{23}$$

Note that, the condition $y \neq x, -x$ is to guarantee that $c \neq 0, \infty$. We compute $c$ in details as follows:

$$c = \frac{-\langle x, y \rangle}{1 - \langle x, y \rangle} \cdot x + \frac{1}{1 - \langle x, y \rangle} \cdot y, \tag{24}$$

so

$$d = \frac{c}{\|c\|_2} = \frac{\dfrac{-\langle x, y \rangle}{1 - \langle x, y \rangle} \cdot x + \dfrac{1}{1 - \langle x, y \rangle} \cdot y}{\left\| \dfrac{-\langle x, y \rangle}{1 - \langle x, y \rangle} \cdot x + \dfrac{1}{1 - \langle x, y \rangle} \cdot y \right\|_2} \tag{25}$$

$$= \frac{\dfrac{-\langle x, y \rangle}{1 - \langle x, y \rangle} \cdot x + \dfrac{1}{1 - \langle x, y \rangle} \cdot y}{\sqrt{\left\langle \dfrac{-\langle x, y \rangle}{1 - \langle x, y \rangle} \cdot x + \dfrac{1}{1 - \langle x, y \rangle} \cdot y \right\rangle}} \tag{26}$$

$$= \frac{\dfrac{-\langle x,y \rangle}{1-\langle x,y \rangle} \cdot x + \dfrac{1}{1-\langle x,y \rangle} \cdot y}{\sqrt{\|x\|_2^2 \cdot \dfrac{\langle x,y \rangle^2}{(1-\langle x,y \rangle)^2} + \|y\|_2^2 \cdot \dfrac{1}{(1-\langle x,y \rangle)^2} - 2 \cdot \langle x,y \rangle \cdot \dfrac{\langle x,y \rangle}{(1-\langle x,y \rangle)^2}}} \tag{27}$$

$$= \frac{\dfrac{-\langle x,y \rangle}{1-\langle x,y \rangle} \cdot x + \dfrac{1}{1-\langle x,y \rangle} \cdot y}{\sqrt{\dfrac{1-\langle x,y \rangle^2}{(1-\langle x,y \rangle)^2}}} \tag{28}$$

$$= \frac{-\langle x,y \rangle \cdot x + y}{\sqrt{1-\langle x,y \rangle^2}}. \tag{29}$$

Note that, since $b$ is the projection of $y$ on vector $x$, so we have

$$b = \langle x,y \rangle \cdot x. \tag{30}$$

By similarity, we have

$$\frac{\text{length of arc from } y \text{ to } a \text{ on the circle centered at } b \text{ passes through } y \text{ and } a}{\text{length of arc from } d \text{ to } y_i \text{ on the circle centered at } 0 \text{ passes through } d \text{ and } y_i} = \frac{\|y-b\|_2}{\|d-0\|_2}. \tag{31}$$

Note that, the length of arc from $d$ to $y_i$ on the circle centered at $0$ passes through $d$ and $y_i$ is

$$d_{\mathbb{S}^d}(d, y_i) = \arccos \langle d, y_i \rangle, \tag{32}$$

so

$$\text{length of arc from } y \text{ to } a \text{ on the circle centered at } b \text{ passes through } y \text{ and } a \tag{33}$$

$$= \arccos \langle d, y_i \rangle \cdot \frac{\|y-b\|_2}{\|d-0\|_2} \tag{34}$$

$$= \arccos \langle d, y_i \rangle \cdot \|y-b\|_2 \tag{35}$$

$$= \arccos \left\langle \frac{-\langle x,y \rangle \cdot x + y}{\sqrt{1-\langle x,y \rangle^2}}, y_i \right\rangle \cdot \|y - \langle x,y \rangle \cdot x\|_2 \tag{36}$$

$$= \arccos \left( \frac{\langle y, y_i \rangle}{\sqrt{1-\langle x,y \rangle^2}} \right) \cdot \sqrt{\|y\|_2^2 + \langle x,y \rangle^2 \cdot \|x\|_2^2 - 2 \langle x,y \rangle \cdot \langle x,y \rangle} \tag{37}$$

$$= \arccos \left( \frac{\langle y, y_i \rangle}{\sqrt{1-\langle x,y \rangle^2}} \right) \cdot \sqrt{1-\langle x,y \rangle^2} \tag{38}$$

$$= \beta(y, \mathcal{T}^x_{y_1,\dots,y_k})_i. \tag{39}$$

We finish the derivation of $\beta$. In context of splitting maps, this is a reasonable choice, since it relates to evaluate distances from a point to a spherical ray.

**2.** The derivation of $\beta$ clearly implies that $\beta$ is continuous. We can also check the continuous of $\beta$ directly from the formula of $\beta$. Since $\beta$ is continuous, we have $\alpha$ is continuous.

**3.** We have $\beta$ is $\mathrm{O}(d+1)$-invariant since orthogonal transformations preserve the standard dot product. Since $\beta$ is $\mathrm{O}(d+1)$-invariant, we have $\alpha$ is $\mathrm{O}(d+1)$-invariant. $\qquad \square$

### A.3 Proof of Theorem 4.3

*Proof.* Recall the notion of (vertical) Radon Transform (Quellmalz et al., 2023). Let $\Phi^d$ be the collection of all spherical rays on $\mathbb{S}^d$, i.e.

$$\Phi^d := \{r_y^x : \ x \in \mathbb{S}^d, y \in H_x\}. \tag{40}$$

Note that, this is the same as the collection of all spherical trees with one edge, i.e. $\mathbb{T}_1^d$. For $f \in L^1(\mathbb{S}^d)$, consider the map $\left(\mathcal{R}r_y^x\right)f \colon r_y^x \equiv [0, \pi] \to \mathbb{R}$ defined by

$$\left(\mathcal{R}r_y^x\right)f(t) = \int_{\mathbb{S}^d} f(z) \cdot \delta(t - \arccos\langle z, x\rangle)dz. \tag{41}$$

Similar to Appendix A.1, we can show that $\left(\mathcal{R}r_y^x\right)f \in L^1(r_y^x)$. We have an operator

$$\mathcal{R} \colon L^1(\mathbb{S}^d) \longrightarrow \bigsqcup_{r_y^x \in \Phi^d} L^1(r_y^x) \tag{42}$$

$$f \longmapsto \left(\left(\mathcal{R}r_y^x\right)f\right)_{r_y^x \in \Phi^d} \tag{43}$$

This is exactly the (vertical) Radon Transform for Lebesgue integrable functions on $\mathbb{S}^d$, as in (Quellmalz et al., 2023). This is proved to be an injective linear operator, so if $\left(\mathcal{R}r_y^x\right)f = 0$ for all $r_y^x \in \Phi^d$, then $f = 0$.

Back to the problem. Recall that $\mathbb{T}_k^d$ is the space of all spherical trees of $k$ edges on $\mathbb{S}^d$,

$$\mathbb{T}_k^d = \{\mathcal{T}_{y_1,\ldots,y_j}^x = (r_{y_1}^x, \ldots r_{y_k}^x) \ : \ x \in \mathbb{S}^d \text{ and } y_1, \ldots, y_k \in H_x\}. \tag{44}$$

For an $1 \leqslant i \leqslant k$ and $r_y^x \in \Phi^d$, define

$$\mathbb{T}_k^d(i, r_y^x) := \left\{\mathcal{T}_{y_1,\ldots,y_j}^x \ : \ y = y_i\right\}. \tag{45}$$

In words, $\mathbb{T}_k^d(i, r_y^x)$ is a subcollection of $\mathbb{T}_k^d$ consists of all spherical trees with root $x$ and the $i^{\text{th}}$ spherical ray is $r_{y_i}^x$. It is clear that $\mathbb{T}_k^d$ is the disjoint union of all $\mathbb{T}_k^d(i, r_y^x)$ for $r_y^x \in \Phi^d$,

$$\mathbb{T}_k^d = \bigsqcup_{r_y^x \in \Phi^d} \mathbb{T}_k^d(i, r_y^x). \tag{46}$$

We have some observations on subcollections $\mathbb{T}_k^d(i, r_y^x)$.

**Result 1.** Each orthogonal transformation $g \in \mathrm{O}(d + 1)$ define a bijection between $\mathbb{T}_k^d(i, r_y^x)$ and $\mathbb{T}_k^d(i, r_{gy}^{gx})$. In details, the map $\phi_g$ defined by

$$\phi_g \colon \quad \mathbb{T}_k^d(i, r_y^x) \quad \longrightarrow \quad \mathbb{T}_k^d(i, r_{gy}^{gx}) \tag{47}$$

$$\mathcal{T}_{y_1,\ldots,y_k}^x \longmapsto \mathcal{T}_{gy_1,\ldots,gy_k}^{gx}. \tag{48}$$

is a well-defined and is a bijection. This can be verified directly by definitions.

**Result 2.** For $1 \leqslant i \leqslant k$ and $r_y^x, r_{y'}^{x'} \in \Phi^d$, we have

$$\int_{\mathbb{T}_k^d(i, r_y^x)} \alpha(z, \mathcal{T})_i \, d\mathcal{T} = \int_{\mathbb{T}_k^d(i, r_{y'}^{x'})} \alpha(z', \mathcal{T})_i \, d\mathcal{T} \tag{49}$$

for all $z, z' \in \mathbb{S}^d$ such that $d_{\mathbb{S}^d}(x, z) = d_{\mathbb{S}^d}(x', z')$. Note that, the intergrations are taken over a $\mathbb{T}_k^d(i, r_y^x)$ and $\mathbb{T}_k^d(i, r_{y'}^{x'})$ with measures induced from the measure of $\mathbb{L}_k^d$. To prove Equation (49), we first show it in two specific cases as follows:

- Case 1. Assume $x = x$ and $y = y$.

- Case 2. Assume $z$ lies on $r_y^x$ and $z'$ lies on $r_{y'}^{x'}$.

If we can show that Equation (49) holds for assumptions in case 1 and 2, then Equation (49) holds for all $x, y, z, x', y', z'$. Indeed, assume that Equation (49) holds for assumptions in case 1 and 2. Then for all $x, y, z, x', y', z'$, we consider $t \in r_y^x$ and $t' \in r_{y'}^{x'}$ such that

$$d_{\mathbb{S}^d}(x, t) = d_{\mathbb{S}^d}(x, z) = d_{\mathbb{S}^d}(x', z') = d_{\mathbb{S}^d}(x', t'). \tag{50}$$

Then from the results in case 1 and 2, we have

$$\int_{\mathbb{T}_k^d(i, r_y^x)} \alpha(z, \mathcal{T})_i \, d\mathcal{T} \stackrel{\text{by case 1}}{=} \int_{\mathbb{T}_k^d(i, r_y^x)} \alpha(t, \mathcal{T})_i \, d\mathcal{T} \tag{51}$$

$$\stackrel{\text{by case 2}}{=} \int_{\mathbb{T}_k^d(i, r_{y'}^{x'})} \alpha(t', \mathcal{T})_i \, d\mathcal{T} \stackrel{\text{by case 1}}{=} \int_{\mathbb{T}_k^d(i, r_{y'}^{x'})} \alpha(z', \mathcal{T})_i \, d\mathcal{T}. \tag{52}$$

So Equation (49) holds for all $x, y, z, x', y', z'$. Now we prove it holds for case 1 and 2.

For case 1, from the transitivity of orthogonal transformations on $\mathbb{S}^d$, there exists $g \in \mathrm{O}(d+1)$ such that

$$gx = x, gy = y, gz = z'. \tag{53}$$

From **Result 1**, there is a corresponding bijection $\phi_g$ from $\mathbb{T}_k^d(i, r_y^x)$ to $\mathbb{T}_k^d(i, r_y^x)$. We have

$$\int_{\mathbb{T}_k^d(i, r_y^x)} \alpha(z', \mathcal{T})_i \, d\mathcal{T} = \int_{\mathbb{T}_k^d(i, r_y^x)} \alpha(z', g\mathcal{T})_i \, d(g\mathcal{T}) \quad \text{(change of variables)} \tag{54}$$

$$= \int_{\mathbb{T}_k^d(i, r_y^x)} \alpha(gz, g\mathcal{T})_i \, d(g\mathcal{T}) \quad \text{(since } z' = gz\text{)} \tag{55}$$

$$= \int_{\mathbb{T}_k^d(i, r_y^x)} \alpha(z, \mathcal{T})_i \, d(g\mathcal{T}) \quad \text{(since } \alpha \text{ is } \mathrm{O}(d+1)\text{-invariant)} \tag{56}$$

$$= \int_{\mathbb{T}_k^d(i, r_y^x)} \alpha(z, \mathcal{T})_i \, d(\mathcal{T}) \quad \text{(since } |\det(g)| = 1\text{)} \tag{57}$$

$$\tag{58}$$

So Equation (49) holds for case 1. A similar proof can be processed for case 2. From the transitivity of orthogonal transformations on $\mathbb{S}^d$, there exists $h \in \mathrm{O}(d+1)$ such that

$$hx = x', hy = y', hz = z'. \tag{59}$$

From **Result 1**, there is a corresponding bijection $\phi_h$ from $\mathbb{T}_k^d(i, r_y^x)$ to $\mathbb{T}_k^d(i, r_{y'}^{x'})$. We have

$$\int_{\mathbb{T}_k^d(i, r_{y'}^{x'})} \alpha(z', \mathcal{T})_i \, d\mathcal{T} = \int_{\mathbb{T}_k^d(i, r_y^x)} \alpha(z', h\mathcal{T})_i \, d(h\mathcal{T}) \quad \text{(change of variables)} \tag{60}$$

$$= \int_{\mathbb{T}_k^d(i, r_y^x)} \alpha(hz, h\mathcal{T})_i \, d(h\mathcal{T}) \quad \text{(since } z' = hz\text{)} \tag{61}$$

$$= \int_{\mathbb{T}_k^d(i, r_y^x)} \alpha(z, \mathcal{T})_i \, d(h\mathcal{T}) \quad \text{(since } \alpha \text{ is } \mathrm{O}(d+1)\text{-invariant)} \tag{62}$$

$$= \int_{\mathbb{T}_k^d(i, r_y^x)} \alpha(z, \mathcal{T})_i \, d(\mathcal{T}) \quad \text{(since } |\det(h)| = 1\text{)} \tag{63}$$

$$\tag{64}$$

We finish the proof for **Result 2**.

**Result 3.** From **Result 2**, for all $1 \leqslant i \leqslant k$ and $t \in [0, \pi]$, we can define a constant $c_i(t)$ such that

$$c_i(t) := \int_{\mathbb{T}_k^d(i, r_y^x)} \alpha(z, \mathcal{T})_i \, d\mathcal{T} \tag{65}$$

for all $r_y^x \in \Phi^d$ and $z \in \mathbb{S}^d$ such that $t = d_{\mathbb{S}^d}(x, z) = \arccos \langle x, z \rangle$. Then for all $t \in [0, \pi]$, we have

$$c_1(t) + c_2(t) + \ldots + c_k(t) = 1. \tag{66}$$

To show this, first, denote $\mathbb{T}_k^d(x)$ as the collection of all spherical trees with root $x$ on $\mathbb{S}^d$. We have

$$\mathbb{T}_k^d(x) = \bigsqcup_{y \in H_x \cap \mathbb{S}^d} \mathbb{T}_k^d(i, r_y^x), \tag{67}$$

so we have

$$\int_{\mathbb{T}_k^d(x)} \alpha(z, \mathcal{T})_i \, d\mathcal{T} = \int_{H_x \cap \mathbb{S}^d} \left( \int_{\mathbb{T}_k^d(i, r_y^x)} \alpha(z, \mathcal{T})_i \, d\mathcal{T} \right) dy \tag{68}$$

$$= \int_{H_x \cap \mathbb{S}^d} c_i(\arccos \langle x, z \rangle) \, dy \tag{69}$$

$$= c_i(\arccos \langle x, z \rangle). \tag{70}$$

Then

$$c_1(\arccos \langle x, z \rangle) + \ldots + c_k(\arccos \langle x, z \rangle) = \sum_{i=1}^{k} \int_{\mathbb{T}_k^d(x)} \alpha(z, \mathcal{T})_i \, d\mathcal{T} \tag{71}$$

$$= \int_{\mathbb{T}_k^d(x)} \left( \sum_{i=1}^{k} \alpha(z, \mathcal{T})_i \right) d\mathcal{T} \tag{72}$$

$$= \int_{\mathbb{T}_k^d(x)} 1 \, d\mathcal{T} \tag{73}$$

$$= 1. \tag{74}$$

We finish the proof for **Result 3**.

Consider a splitting map $\alpha$ in $\mathcal{C}(\mathbb{S}^d \times \mathbb{T}_k^d, \Delta_{k-1})$ that is $O(d+1)$-invariant. For a function $f \in L^1(\mathbb{S}^d)$, for each $1 \leqslant i \leqslant k$, define a function $g_i \in L^1([0, \pi] \times \Phi^d)$ as follows

$$g_i : \quad [0, \pi] \times \Phi^d \quad \longrightarrow \quad \mathbb{R} \tag{75}$$

$$(t, r_y^x) \quad \longmapsto \quad \int_{\mathbb{T}_k^d(i, r_y^x)} \mathcal{R}_{\mathcal{T}}^\alpha f(t, r_y^x) \, d\mathcal{T}. \tag{76}$$

From the definition of $\mathcal{R}_{\mathcal{T}}^\alpha f$,

$$\mathcal{R}_{\mathcal{T}}^\alpha f : \quad \mathcal{T} \quad \longrightarrow \mathbb{R} \tag{77}$$

$$(t, r_{y_i}^x) \longmapsto \int_{\mathbb{S}^d} f(y) \cdot \alpha(y, \mathcal{T})_i \cdot \delta(t - \arccos \langle x, y \rangle) \, dy, \tag{78}$$

we have

$$g_i(t, r_y^x) = \int_{\mathbb{T}_k^d(i, r_y^x)} \mathcal{R}_{\mathcal{T}}^\alpha f(t, r_y^x) \, d\mathcal{T} \tag{79}$$

$$= \int_{\mathbb{T}_k^d(i, r_y^x)} \left( \int_{\mathbb{S}^d} f(z) \cdot \alpha(z, \mathcal{T})_i \cdot \delta(t - \arccos \langle x, z \rangle) \, dz \right) d\mathcal{T} \tag{80}$$

$$= \int_{\mathbb{S}^d} \left( \int_{\mathbb{T}_k^d(i, r_y^x)} f(z) \cdot \alpha(z, \mathcal{T})_i \cdot \delta(t - \arccos \langle x, z \rangle) \, d\mathcal{T} \right) dz \tag{81}$$

$$= \int_{\mathbb{S}^d} f(z) \cdot \delta(t - \arccos \langle x, z \rangle) \cdot \left( \int_{\mathbb{T}_k^d(i, r_y^x)} \alpha(z, \mathcal{T})_i \, d\mathcal{T} \right) dz \tag{82}$$

$$= \int_{\mathbb{S}^d} f(z) \cdot \delta(t - \arccos \langle x, z \rangle) \cdot c_i(\arccos \langle x, z \rangle) \, dz \tag{83}$$

$$= c_i(t) \cdot \int_{\mathbb{S}^d} f(z) \cdot \delta(t - \arccos \langle x, z \rangle) \, dz \tag{84}$$

$$= c_i(t) \cdot \left( \mathcal{R} r_y^x \right) f(t). \tag{85}$$

So

$$\sum_{i=1}^{k} g_i(t, r_y^x) = \sum_{i=1}^{k} c_i(t) \cdot \left( \mathcal{R} r_y^x \right) f(t) \tag{86}$$

$$= \left( \sum_{i=1}^{k} c_i(t) \right) \cdot \left( \mathcal{R} r_y^x \right) f(t) = 1 \cdot \left( \mathcal{R} r_y^x \right) f(t) = \left( \mathcal{R} r_y^x \right) f(t) \tag{87}$$

Let $f \in \operatorname{Ker} \mathcal{R}^\alpha$, which means $\mathcal{R}_{\mathcal{T}}^\alpha f = 0$ for all $\mathcal{T} \in \mathbb{T}_k^d$. So $g = 0 \in L^1([0, \pi] \times \Phi^d)$ for all $1 \leqslant i \leqslant k$. It implies $\left( \mathcal{R} r_y^x \right) f = 0 \in L^1(r_y^x)$ for all $r_y^x \in \Phi^d$. So, from the (vertical) Radon Transform is injective, we conclude that $f = 0 \in L^1(\mathbb{S}^d)$. so $\mathcal{R}^\alpha$ is injective. □

*Remark.* To formalize the proof above, the notion of Haar measure for compact groups is required. However, we simplify the explanation as it goes beyond the scope of this paper.

### A.4 PROOF OF THEOREM 5.2

*Proof.* We want to show that

$$\mathrm{STSW}(\mu, \nu) = \int_{\mathbb{T}_k^d} \mathrm{W}_{d_{\mathcal{T}}, 1}(\mathcal{R}_{\mathcal{T}}^\alpha \mu, \mathcal{R}_{\mathcal{T}}^\alpha \nu) \, d\sigma(\mathcal{T}). \tag{88}$$

is a metric on $\mathcal{P}(\mathbb{S}^d)$.

**Positive definiteness.** For $\mu, \nu \in \mathcal{P}(\mathbb{S}^d)$, it is clear that $\mathrm{STSW}(\mu, \mu) = 0$ and $\mathrm{STSW}(\mu, \nu) \geqslant 0$. If $\mathrm{STSW}(\mu, \nu) = 0$, then $\mathrm{W}_{d_{\mathcal{T}}, 1}(\mathcal{R}_{\mathcal{T}}^\alpha \mu, \mathcal{R}_{\mathcal{T}}^\alpha \nu) = 0$ for all $\mathcal{T} \in \mathbb{T}_k^d$. Since $\mathrm{W}_{d_{\mathcal{T}}, 1}$ is a metric on $\mathcal{P}(\mathcal{T})$, we have $\mathcal{R}_{\mathcal{T}}^\alpha \mu = \mathcal{R}_{\mathcal{T}}^\alpha \nu$ for all $\mathcal{T} \in \mathbb{T}_k^d$. By the injectivity of our Radon transform variant, we have $\mu = \nu$.

**Symmetry.** For $\mu, \nu \in \mathcal{P}(\mathbb{S}^d)$, we have:

$$\mathrm{STSW}(\mu, \nu) = \int_{\mathbb{T}_k^d} \mathrm{W}_{d_{\mathcal{T}}, 1}(\mathcal{R}_{\mathcal{T}}^\alpha \mu, \mathcal{R}_{\mathcal{T}}^\alpha \nu) \, d\sigma(\mathcal{T}) \tag{89}$$

$$= \int_{\mathbb{T}_k^d} \mathrm{W}_{d_{\mathcal{T}}, 1}(\mathcal{R}_{\mathcal{T}}^\alpha \nu, \mathcal{R}_{\mathcal{T}}^\alpha \mu) \, d\sigma(\mathcal{T}) = \mathrm{STSW}(\nu, \mu). \tag{90}$$

So $\mathrm{STSW}(\mu, \nu) = \mathrm{STSW}(\nu, \mu)$.

**Triangle inequality.** For $\mu_1, \mu_2, \mu_3 \in \mathcal{P}(\mathbb{S}^d)$, we have:

$$\mathrm{STSW}(\mu_1, \mu_2) + \mathrm{STSW}(\mu_2, \mu_3) \tag{91}$$

$$= \int_{\mathbb{T}_k^d} \mathrm{W}_{d_{\mathcal{T}}, 1}(\mathcal{R}_{\mathcal{T}}^\alpha \mu_1, \mathcal{R}_{\mathcal{T}}^\alpha \mu_2) \, d\sigma(\mathcal{T}) + \int_{\mathbb{T}_k^d} \mathrm{W}_{d_{\mathcal{T}}, 1}(\mathcal{R}_{\mathcal{T}}^\alpha \mu_2, \mathcal{R}_{\mathcal{T}}^\alpha \mu_3) \, d\sigma(\mathcal{T}) \tag{92}$$

$$= \int_{\mathbb{T}_k^d} \left( \mathrm{W}_{d_{\mathcal{T}}, 1}(\mathcal{R}_{\mathcal{T}}^\alpha \mu_1, \mathcal{R}_{\mathcal{T}}^\alpha \mu_2) + \mathrm{W}_{d_{\mathcal{T}}, 1}(\mathcal{R}_{\mathcal{T}}^\alpha \mu_2, \mathcal{R}_{\mathcal{T}}^\alpha \mu_3) \right) d\sigma(\mathcal{T}) \tag{93}$$

$$\geqslant \int_{\mathbb{T}_k^d} \mathrm{W}_{d_{\mathcal{T}}, 1}(\mathcal{R}_{\mathcal{T}}^\alpha \mu_1, \mathcal{R}_{\mathcal{T}}^\alpha \mu_3) \, d\sigma(\mathcal{T}) \tag{94}$$

$$= \mathrm{STSW}(\mu_1, \mu_3). \tag{95}$$

So the triangle inequality holds for STSW.

We conclude that STSW is a metric on the space $\mathcal{P}(\mathbb{S}^d)$.

$\mathrm{O}(d+1)$-**invariance of** STSW. For $g \in \mathrm{O}(d+1)$, we show that

$$\mathrm{STSW}(\mu, \nu) = \mathrm{STSW}(g\sharp\mu, g\sharp\nu), \tag{96}$$

where $g\sharp\mu, g\sharp\nu$ as the pushforward of $\mu, \nu$ via orthogonal transformation $g \colon \mathbb{S}^d \to \mathbb{S}^d$, respectively. For $\mathcal{T} = \mathcal{T}^x_{y_1,\dots,y_k} \in \mathbb{T}^d_k$, we have $g\mathcal{T} = \mathcal{T}^{gx}_{gy_1,\dots,gy_k}$. Note that $|\det(g)| = 1$, so

$$\mathcal{R}^\alpha_{g\mathcal{T}}(g\sharp\mu)(t, r^{gx}_{gy_i}) = \int_{\mathbb{S}^d} g\sharp\mu(y) \cdot \alpha(y, g\mathcal{T})_i \cdot \delta(t - \arccos\langle gx, y\rangle)\, dy \tag{97}$$

$$= \int_{\mathbb{S}^d} \mu(g^{-1}y) \cdot \alpha(y, g\mathcal{T})_i \cdot \delta(t - \arccos\langle gx, y\rangle)\, dy \tag{98}$$

$$= \int_{\mathbb{S}^d} \mu(g^{-1}gy) \cdot \alpha(gy, g\mathcal{T})_i \cdot \delta(t - \arccos\langle gx, gy\rangle)\, d(gy) \tag{99}$$

$$= \int_{\mathbb{S}^d} \mu(y) \cdot \alpha(y, \mathcal{T})_i \cdot \delta(t - \arccos\langle x, y\rangle)\, d(y) \tag{100}$$

$$= \mathcal{R}^\alpha_{\mathcal{T}}\mu(t, r^x_{y_i}). \tag{101}$$

Similarly, we have

$$\mathcal{R}^\alpha_{g\mathcal{T}}(g\sharp\nu)(t, r^{gx}_{gy_i}) = \mathcal{R}^\alpha_{\mathcal{T}}\nu(t, r^x_{y_i}). \tag{102}$$

Since $g$ induces an isometric transformation $\mathcal{T} \to g\mathcal{T}$, so

$$\mathrm{W}_{d_{\mathcal{T}},1}(\mathcal{R}^\alpha_{\mathcal{T}}\mu, \mathcal{R}^\alpha_{\mathcal{T}}\nu) = \mathrm{W}_{d_{g\mathcal{T}},1}(\mathcal{R}^\alpha_{g\mathcal{T}}g\sharp\mu, \mathcal{R}^\alpha_{g\mathcal{T}}g\sharp\nu). \tag{103}$$

We have

$$\mathrm{STSW}(g\sharp\mu, g\sharp\nu) = \int_{\mathbb{T}^d_k} \mathrm{W}_{d_{\mathcal{T}},1}(\mathcal{R}^\alpha_{\mathcal{T}}g\sharp\mu, \mathcal{R}^\alpha_{\mathcal{T}}g\sharp\nu)\, d\sigma(\mathcal{T}) \tag{104}$$

$$= \int_{\mathbb{T}^d_k} \mathrm{W}_{d_{g\mathcal{T}},1}(\mathcal{R}^\alpha_{g\mathcal{T}}g\sharp\mu, \mathcal{R}^\alpha_{g\mathcal{T}}g\sharp\nu)\, d\sigma(g\mathcal{T}) \tag{105}$$

$$= \int_{\mathbb{T}^d_k} \mathrm{W}_{d_{\mathcal{T}},1}(\mathcal{R}^\alpha_{\mathcal{T}}\mu, \mathcal{R}^\alpha_{\mathcal{T}}\nu)\, d\sigma(\mathcal{T}) \tag{106}$$

$$= \mathrm{STSW}(\mu, \nu) \tag{107}$$

So STSW is $\mathrm{O}(d+1)$-invariant. $\qquad\square$

### A.5 DERIVATION FOR THE CLOSED-FORM APPROXIMATION OF STSW

We derive the closed-form approximation of STSW for two discrete probability distributions $\mu$ and $\nu$ given as follows

$$\mu(x) = \sum_{j=1}^n u_j \cdot \delta(x - a_j) \quad \text{and} \quad \nu(x) = \sum_{i=j}^n v_j \cdot \delta(x - a_j). \tag{108}$$

We can write $\mu$ and $\nu$ in these forms by combining their supports and allow some $u_j$ and $v_j$ to be 0. Consider spherical tree $\mathcal{T} = \mathcal{T}^x_{y_1,\dots,y_k}$. For $1 \leqslant j \leqslant n$, let $c_j = d_{\mathbb{S}^d}(x, a_j)$, and also let $c_0 = 0$. By re-indexing, we assume that the sequence $c_0, \dots, c_n$ is increasing,

$$0 = c_0 \leqslant c_1 \leqslant c_2 \leqslant \dots \leqslant c_n. \tag{109}$$

For $0 \leqslant j \leqslant n$ and $1 \leqslant i \leqslant k$, consider all points $x^{(i)}_j = (c_j, r^x_{y_i})$ on the spherical tree $\mathcal{T}$. Since $c_0 = 0$, we have

$$x^{(1)}_0 = x^{(2)}_0 = \dots = x^{(k)}_0 = x, \tag{110}$$

and for $1 \leqslant j \leqslant n$, $x^{(i)}_j$ is exactly the unique intersection between the hyperplane passes through $a_j$ and orthogonal to $x$, and the spherical ray $r^x_{y_i}$. We compute $\mathcal{R}^\alpha_{\mathcal{T}}\mu$: For $t \in [0, \pi]$ and $1 \leqslant i \leqslant k$,

$$\mathcal{R}^\alpha_{\mathcal{T}}\mu(t, r^x_{y_i}) = \int_{\mathbb{S}^d} \mu(y) \cdot \alpha(y, \mathcal{T})_i \cdot \delta(t - \arccos\langle x, y\rangle)\, dy \tag{111}$$

$$= \int_{\mathbb{S}^d} \left( \sum_{j=1}^n u_j \cdot \delta(y - a_j) \right) \cdot \alpha(y, \mathcal{T})_i \cdot \delta(t - \arccos \langle x, y \rangle) \, dy \tag{112}$$

$$= \sum_{j=1}^n u_j \cdot \int_{\mathbb{S}^d} \alpha(y, \mathcal{T})_i \cdot \left( \delta(y - a_j) \cdot \delta(t - \arccos \langle x, y \rangle) \right) dy. \tag{113}$$

So,

1. If $t \notin \{c_1, \ldots, c_n\}$, then $\mathcal{R}^\alpha_\mathcal{T} \mu(t, r^x_{y_i}) = 0$; and,

2. If $t = c_j$ for some $j$, then $\mathcal{R}^\alpha_\mathcal{T} \mu(t, r^x_{y_i}) = \mathcal{R}^\alpha_\mathcal{T} \mu(c_j, r^x_{y_i}) = \mathcal{R}^\alpha_\mathcal{T} \mu(x^{(i)}_j) = \alpha(a_j, \mathcal{T})_i \cdot u_j.$

Similarly, we have

1. If $t \notin \{c_1, \ldots, c_n\}$, then $\mathcal{R}^\alpha_\mathcal{T} \nu(t, r^x_{y_i}) = 0$; and,

2. If $t = c_j$ for some $j$, then $\mathcal{R}^\alpha_\mathcal{T} \nu(t, r^x_{y_i}) = \mathcal{R}^\alpha_\mathcal{T} \nu(c_j, r^x_{y_i}) = \mathcal{R}^\alpha_\mathcal{T} \nu(x^{(i)}_j) = \alpha(a_j, \mathcal{T})_i \cdot v_j.$

For $1 \leqslant j \leqslant n$ and $1 \leqslant i \leqslant k$, let

$$u^{(i)}_j = \alpha(a_j, \mathcal{T})_i \cdot u_j \quad \text{and} \quad v^{(i)}_j = \alpha(a_j, \mathcal{T})_i \cdot v_j. \tag{114}$$

Consider $\mathcal{T}$ as a graph with nodes $x^{(i)}_j$ for $1 \leqslant i \leqslant k, 0 \leqslant j \leqslant n$. Note that $x^{(i)}_0 = x$ for all $i$, and we assign this is the root of $\mathcal{T}$. Two nodes is adjacent is the shortest path on $\mathcal{T}$ does not contain any other nodes. In other words, the set of edges in $\mathcal{T}$ are $e^{(i)}_j = (x^{(i)}_j, x^{(i)}_{j-1})$ for $1 \leqslant i \leqslant k, 1 \leqslant j \leqslant n$, and $e^{(i)}_j = (x^{(i)}_j, x^{(i)}_{j-1})$ has length $c_j - c_{j-1}$. For an edge $e^{(i)}_j$, its further endpoint from the root is $x^{(i)}_j$. Also, for a node $x^{(i)}_j$ with $j > 0$, the corresponding subtree $\Gamma(x^{(i)}_j)$ contains all nodes $x^{(i)}_p$ with $j \leqslant p \leqslant n$. From these above observations, we can see $\mu$ and $\nu$ transform to $\mathcal{R}^\alpha_\mathcal{T} \mu$ and $\mathcal{R}^\alpha_\mathcal{T} \nu$ supported on nodes of $\mathcal{T}$, where the mass at node $x^{(i)}_j$ is $u^{(i)}_j$ and $v^{(i)}_j$, respectively. So, from Equation (3), we have

$$W_{d_\mathcal{T}, 1}(\mathcal{R}^\alpha_\mathcal{T} \mu, \mathcal{R}^\alpha_\mathcal{T} \nu) = \sum_{e \in \mathcal{T}} w_e \cdot \left| \mu(\Gamma(v_e)) - \nu(\Gamma(v_e)) \right| \tag{115}$$

$$= \sum_{i=1}^k \sum_{j=1}^n (c_j - c_{j-1}) \cdot \left| \mu(\Gamma(x^{(i)}_j)) - \nu(\Gamma(x^{(i)}_j)) \right| \tag{116}$$

$$= \sum_{j=1}^n (c_j - c_{j-1}) \cdot \left( \sum_{i=1}^k \left| \mu(\Gamma(x^{(i)}_j)) - \nu(\Gamma(x^{(i)}_j)) \right| \right) \tag{117}$$

$$= \sum_{j=1}^n (c_j - c_{j-1}) \cdot \left( \sum_{i=1}^k \left| \sum_{p=j}^n \mu(x^{(i)}_p) - \sum_{p=j}^n \nu(x^{(i)}_p) \right| \right) \tag{118}$$

$$= \sum_{j=1}^n (c_j - c_{j-1}) \cdot \left( \sum_{i=1}^k \left| \sum_{p=j}^n u^{(i)}_p - \sum_{p=j}^n v^{(i)}_p \right| \right) \tag{119}$$

$$= \sum_{j=1}^n (c_j - c_{j-1}) \cdot \left( \sum_{i=1}^k \left| \sum_{p=j}^n \left( u^{(i)}_p - v^{(i)}_p \right) \right| \right) \tag{120}$$

$$= \sum_{j=1}^n (c_j - c_{j-1}) \cdot \left( \sum_{i=1}^k \left| \sum_{p=j}^n \alpha(a_p, \mathcal{T})_i \cdot (u_p - v_p) \right| \right) \tag{121}$$

$$\tag{122}$$

This is identical to Equation (19). We finish the derivation.

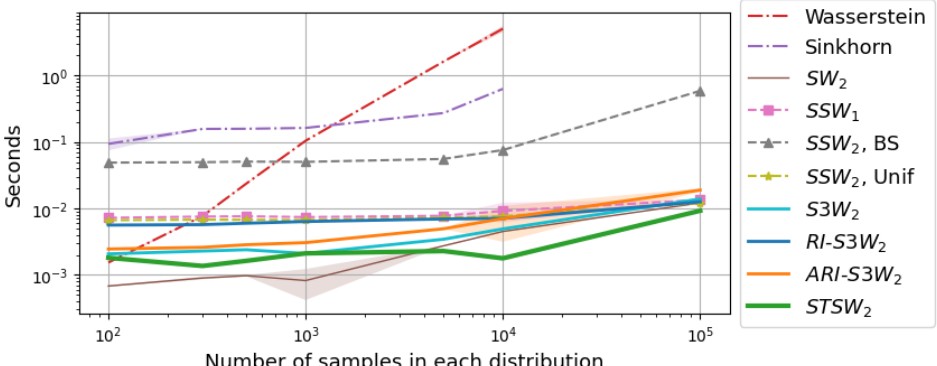

Figure 3: Runtime Comparison, averaged over 15 runs.

## B EXPERIMENTAL DETAILS

All our experiments were conducted on a single NVIDIA H100 80G GPU. For all tasks, if not specified, hyperparameter $\zeta$ in STSW is set to its default value of 2.

### B.1 IMPLEMENTATION

We summarize a pseudo-code for STSW distance computation in Algorithm 1.

---

**Algorithm 1** Spherical Tree-Sliced Wasserstein distance.

---

**Input:** $\mu, \nu \in \mathcal{P}(\mathbb{S}^d)$ as $\mu(x) = \sum_{j=1}^n u_j \cdot \delta(x - a_j)$, $\nu(x) = \sum_{j=1}^n v_j \cdot \delta(x - a_j)$, number of spherical trees $L$, number of rays in spherical trees $k$, splitting maps $\alpha$ with weight $\delta \in \mathbb{R}$.

**for** $l = 1$ to $L$ **do**

    Sample $x^{(l)}, y_1^{(l)}, \ldots, y_k^{(l)} \stackrel{i.i.d}{\sim} \mathcal{N}(0, \mathrm{Id}_{d+1})$.

    Compute $x^{(l)} \leftarrow x^{(l)}/\|x^{(l)}\|_2$ and $y_j^{(l)} \leftarrow \varphi_{x^{(l)}}(y_j^{(l)})/\|\varphi_{x^{(l)}}(y_j^{(l)})\|_2$.

    Contruct spherical tree $\mathcal{T}_l = \mathcal{T}_{y_1^{(l)}, \ldots, y_k^{(l)}}^{x^{(l)}}$.

    Compute $\mathrm{W}_{d_{\mathcal{T}_l}, 1}(\mathcal{R}_{\mathcal{T}_l}^\alpha \mu, \mathcal{R}_{\mathcal{T}_l}^\alpha \nu)$ by Equation (19).

**end for**

Compute $\widehat{\mathrm{STSW}}(\mu, \nu) = (1/L) \cdot \Sigma_{l=1}^L \mathrm{W}_{d_{\mathcal{T}_l}, 1}(\mathcal{R}_{\mathcal{T}_l}^\alpha \mu, \mathcal{R}_{\mathcal{T}_l}^\alpha \nu)$

**Return:** $\widehat{\mathrm{STSW}}(\mu, \nu)$.

---

### B.2 EVOLUTION OF STSW

In this section, we examine the evolution of STSW as well as different distances when measuring two distributions. In line with (Bonet et al., 2022; Tran et al., 2024b), we select source distribution vMF$(\cdot, 0)$ a.k.a uniform distribution and target distribution vMF$(\mu, \kappa)$. We initialize 500 samples in each distribution. We use kappa $\kappa = 10$, $L = 200$ trees, $k = 10$ lines for STSW, $L = 200$ projections for other sliced metrics, $N_R = 100$ rotations for RI-S3W, ARI-S3W, and a pool size of 1000 for ARI-S3W in all experiments unless specified otherwise. Results are averaged over 20 runs.

**Evolution w.r.t $\kappa$.** Figure 4 shows the evolution of various methods w.r.t to $\kappa$. As expected, STSW aligns with the trends in S3W and SSW, decreasing with higher dimensions, unlike KL divergence. Here, we use a derived form for KL divergence (Davidson et al., 2018; Xu & Durrett, 2018) as follows:

$$\mathrm{KL}(\mathrm{vMF}(\mu, \kappa) \| \mathrm{vMF}(\cdot, 0)) = \kappa \frac{I_{(d+1)/2}(\kappa)}{I_{(d+1)/2-1}(\kappa)} + \left(\frac{d+1}{2} - 1\right) \log \kappa - \frac{d+1}{2} \log(2\pi)$$

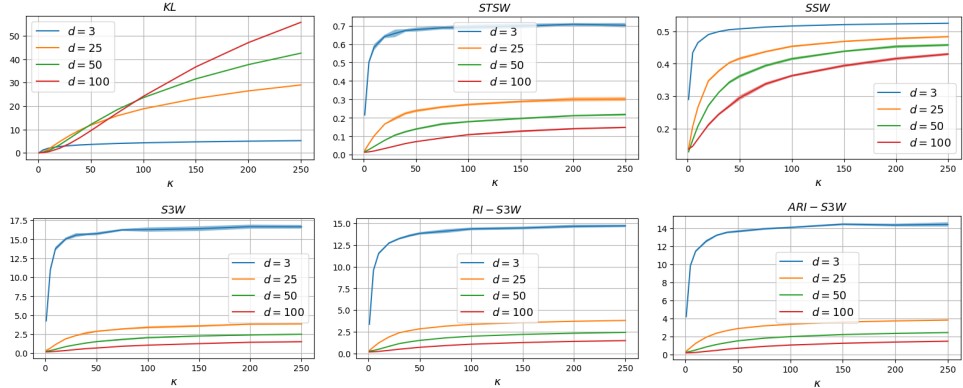

Figure 4: Evolution between vMF($\mu, \kappa$) and vMF($\cdot, 0$)) w.r.t $\kappa$ on $\mathbb{S}^{d-1}$ across various methods. We use $\kappa \in \{1, 5, 10, 20, 30, 40, 50, 75, 100, 150, 200, 250\}$

$$-\log I_{(d+1)/2-1}(\kappa) + \frac{d+1}{2}\log\pi + \log 2 - \log\Gamma\left(\frac{d+1}{2}\right).$$

Figure 5: Evolution between rotated vMFs distributions, averaged over 100 runs. $d$ denotes data dimension.

**Evolution w.r.t rotated vMFs.** Next, we evaluate a fixed vMF distribution and its rotation along a great circle. Specifically, we compute metric between vMF($(1, 0, 0, \dots), \kappa$) and vMF($(\cos\theta, \sin\theta, 0, \dots), \kappa$) for $\theta \in \{(k\pi)/6\}_{k=0}^{12}$. We plot results in Figure 5

**Evolution of STSW w.r.t Number of Trees, Number of Lines and $\zeta$.** Next, we study the effect of the number of trees and lines on STSW. If not specified, we fix $\kappa = 10$ and $d = 3$. We present the results in Figure 6, Figure 7, and Figure 8

### B.3 RUNTIME ANALYSIS

**Runtime Comparison.** We now perform a runtime comparison with other commonly used distance metrics, including the traditional Wasserstein, Sinkhorn (Cuturi, 2013), Sliced-Wasserstain (SW), Spherical Sliced-Wasserstein (SSW) (Bonet et al., 2022) as well as Stereographic Spherical Sliced Wasserstein (S3W) (Tran et al., 2024b) and its variants (RI-S3W, ARI-S3W). For a fair comparison, we also include SSW$_2$ with binary search (BS) and Unif when a closed form is available for uniform distribution. We set $L = 200$ projections for all methods. For our STSW, we use $L = 100$

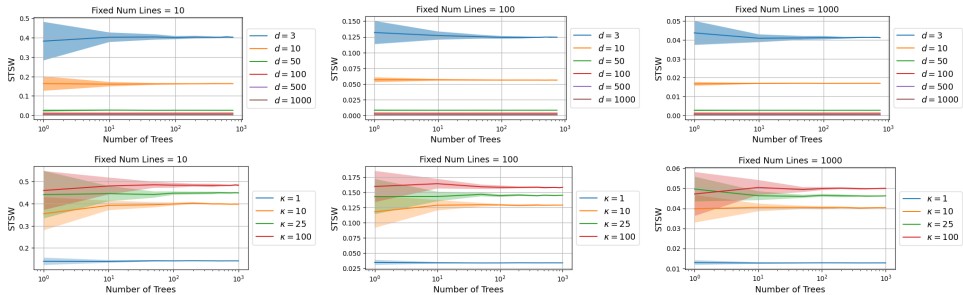

Figure 6: Evolution of STSW between the source and target distributions when varying number of trees $L \in \{1, 10, 50, 100, 200, 400, 500, 750, 900, 1000\}$.

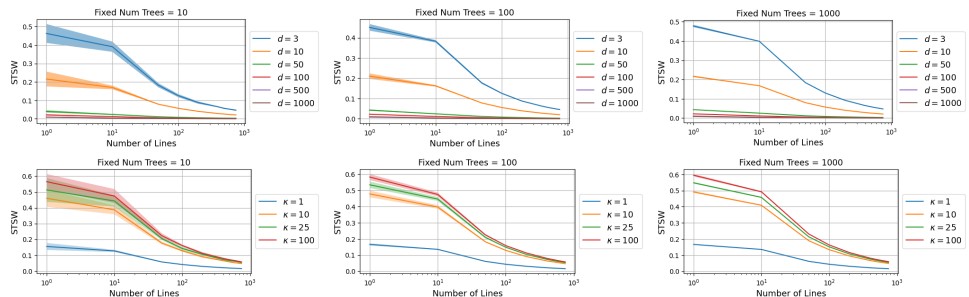

Figure 7: Evolution of STSW between two distributions w.r.t number of lines $k \in \{1, 10, 50, 100, 200, 400, 500, 600, 700, 750\}$.

trees and $k = 10$ lines. The runtime of applying each of these methods on two distribution on $\mathbb{S}^2$ is illustrated in Figure 3.

**Runtime Evolution.** To further assess STSW performance, we conduct a runtime analysis to understand the computational cost associated with different configurations. We again choose uniform distribution and $\text{vMF}(\mu, \kappa)$ where $\kappa = 10$ as our source and target distribution and use STSW to measure distance between these two probabilities. All experiments are repeated 20 times with default parameters set to $L = 200$ trees, $k = 10$ lines and $N = 500$ samples, unless otherwise stated.

We vary the number of trees $L \in \{200, 400, 500, 750, 900, 1000, 1250, 1500, 1750, 2000\}$ in Figure 9a, adjust the number of lines $k$ across $\{5, 10, 25, 50, 100, 150, 200, 300, 500, 750, 1000\}$ in Figure 9b and change the number of samples $N$ within $\{500, 1000, 3000, 5000, 7000, 8000, 10000\}$ in Figure 9c. We note that the runtime of STSW scales linearly with these parameters.

## B.4 GRADIENT FLOW

The probability density function of the von Mises-Fisher distribution with mean direction $\mu \in \mathbb{S}^d$ is given by:

$$f(x; \mu, \kappa) = C_d(\kappa) \exp(\kappa \mu^T x)$$

where $\kappa > 0$ is concentration parameter and the normalization constant $C_d(\kappa) = \dfrac{\kappa^{d/2-1}}{(2\pi)^{p/2} I_{p/2-1}(\kappa)}$

Our target distribution, 12 vMFs with 2400 samples (200 per vFM), have $\kappa = 50$ and

$$\begin{array}{llll} \mu_1 = (-1, \phi, 0), & \mu_2 = (1, \phi, 0), & \mu_3 = (-1, -\phi, 0), & \mu_4 = (1, -\phi, 0) \\ \mu_5 = (0, -1, \phi), & \mu_6 = (0, 1, \phi), & \mu_6 = (0, -1, -\phi), & \mu_8 = (0, 1, -\phi) \\ \mu_9 = (\phi, 0, -1), & \mu_{10} = (\phi, 0, 1), & \mu_{11} = (-\phi, 0, -1), & \mu_{12} = (-\phi, 0, 1) \end{array}$$

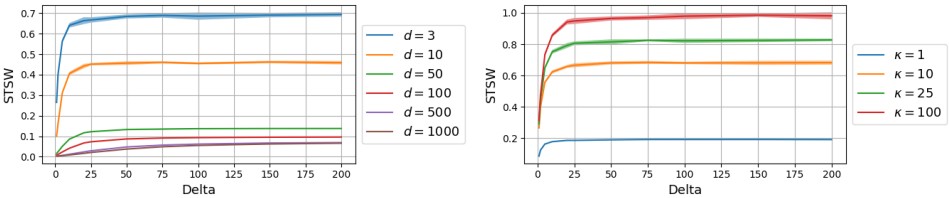

Figure 8: Evolution of STSW between two distributions w.r.t $\zeta \in \{1, 2, 5, 10, 20, 25, 50, 75, 100, 150, 200\}$

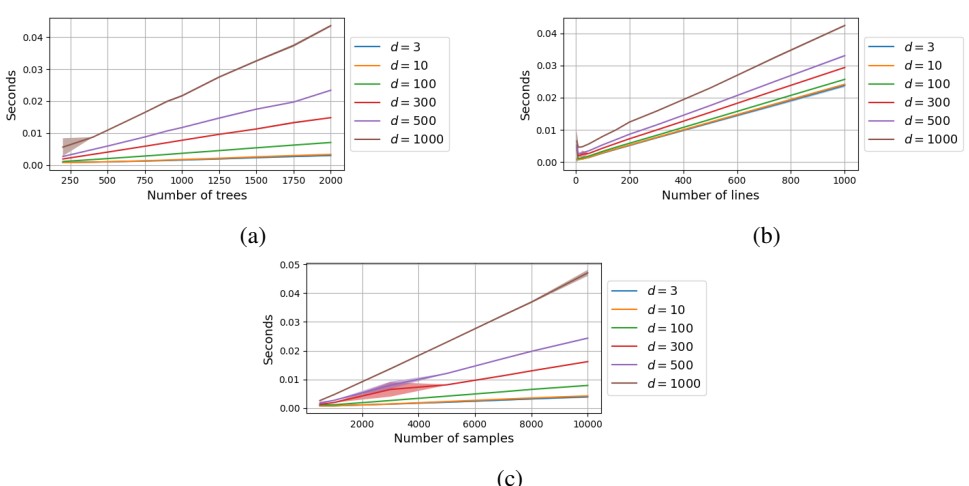

(a)

(b)

(c)

Figure 9: Runtime of STSW w.r.t number of trees, lines and samples

where $\phi = \dfrac{1 + \sqrt{5}}{2}$. The projected gradient descent as described in (Bonet et al., 2022):

$$\begin{cases} x^{(k+1)} = x^{(k)} - \gamma \nabla_{x^{(k)}} \text{STSW}(\hat{\mu}_k, \nu), \\ x^{(k+1)} = \dfrac{x^{(k+1)}}{\|x^{(k+1)}\|_2}, \end{cases}$$

**Setup.** We fix $L = 200$ trees and $k = 5$ lines. For the rest, we use $L = 1000$ projections. As in the original setup, ARI-S3W (30) has 30 rotations with a pool size of 1000 while RI-S3W (1) and RI-S3W (5) have 1 and 5 rotations respectively. We train with Adam (Kinga et al., 2015) optimizer $lr = 0.01$ over 500 epochs and an additional $lr = 0.05$ for SSW.

**Results.** As seen from Table 1 and Figure 10, STSW provides better results in log 2-Wasserstein distance and NLL, while also being efficient in terms of both runtime and convergence speed.

We perform additional experiments on the most informative sliced methods including MAX-STSW, MAX-SSW, and MAX-SW. We present in Table 5 the results after training for 1000 epochs with a learning rate $LR = 0.01$. Each experiment is repeated 10 times. Figure 11 illustrates the log 2-Wasserstein distance between the source and target distribution during training. We observe that MAX-STSW performs better than others.

## B.5 SELF-SUPERVISED LEARNING

**Encoder.** Consistent with the setup in (Bonet et al., 2022; Tran et al., 2024b), we train a ResNet18 (He et al., 2016) on CIFAR-10 (Krizhevsky, 2009) data for 200 epochs using a batch size of 512. We use SGD as our optimizer with initial $lr = 0.05$ a momentum 0.9, and a weight decay $10^{-3}$. The standard data augmentations used to generate positive pairs are similar to prior works (Wang & Isola, 2020; Bonet et al., 2022; Tran et al., 2024b) which include resizing, cropping, horizontal flipping, color jittering, and random grayscale conversion.

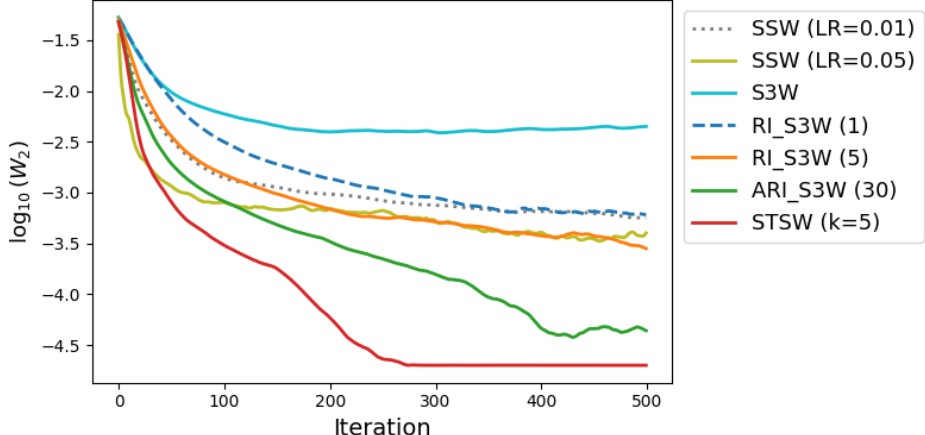

Figure 10: Log 2-Wasserstein distance between source and target distributions

Table 5: Learning target distribution 12 vFMs, trained for 1000 epochs and averaged over 10 runs.

| Method | $\log W_2 \downarrow$ | NLL $\downarrow$ |
|---|---|---|
| MAX-SW | $-3.10 \pm 0.06$ | $-4959.14 \pm 12.22$ |
| MAX-SSW | $-2.76 \pm 0.02$ | $-4868.78 \pm 60.51$ |
| MAX-STSW | $\mathbf{-3.19 \pm 0.03}$ | $\mathbf{-5007.72 \pm 16.34}$ |

We set $L = 200$ trees and $k = 20$ lines for STSW and fix $L = 200$ projections for all other sliced distances. $N_R = 5$ and a pool size of 100 are used for RI-S3W and ARI-S3W as in Tran et al. (2024b). For settings of the regularization coefficient, please refer to Table 6.

**Linear Classifier.** A linear classifier is then trained on feature representations from the pre-trained encoder. Similar to Bonet et al. (2022), we train it for 100 epochs using the Adam (Kinga et al., 2015) optimizer with a learning rate of $10^{-3}$, a weight decay of $0.2$ at epoch 60 and 80.

**Results.** We report in Table 2 the best accuracy of the linear evaluation on features taken before and after projection on $\mathbb{S}^d$ where $d = 9$. The visualizations of learned representations when $d = 2$ can be found in Figure 12.

### B.6  EARTH DATA ESTIMATION

Similar to Bonet et al. (2022) and Tran et al. (2024b), we use an exponential mapping normalizing flows model consisting of 48 radial blocks with 100 components each, totaling 24000 parameters. The model is then trained with full batch gradient descent via Adam optimizer. Dataset details are provided in Table 7.

**Setup.** Our settings for STSW in this task are $L = 1000$ trees, $k = 100$ lines, and $\zeta = -100$. We use $lr = 0.05$ for STSW, S3W, RI-S3W and ARI-S3W and $lr = 0.1$ for SW and SSW. We train other sliced distances for 20,000 epochs as in the original setup while our STSW is only trained for 10,000 epochs.

**Results.** Table 3 highlights the competitive performance of STSW compared to the baseline methods. To further evaluate the efficiency of our approach, we compare the training time of STSW with that of the second-best performer, ARI-S3W, using the Fire dataset. Our findings show that STSW (2 hours 10 minutes) is twice as fast as ARI-S3W (4 hours 30 minutes). We also present in Figure 13 the normalized density maps of test data learned.

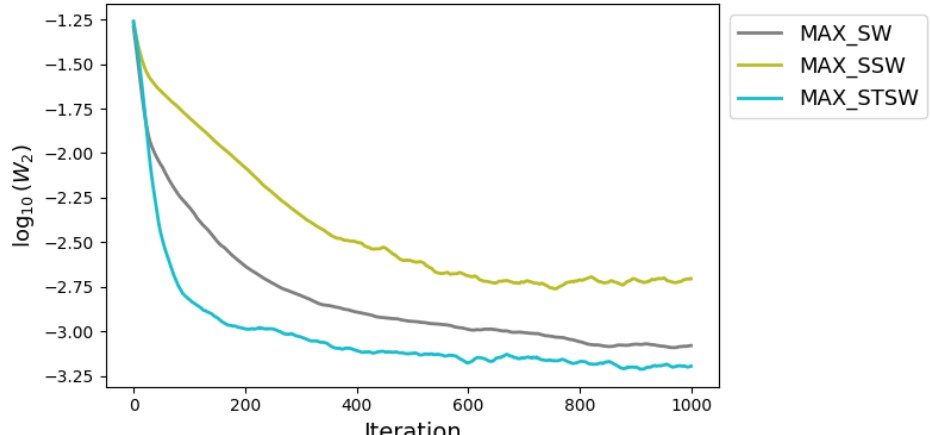

Figure 11: Log 2-Wasserstein distance between source and target distributions.

Table 6: Regularization coefficient $\lambda$ across various methods w.r.t projection on $\mathbb{S}^d$ in Self-Supervised Learning task.

|  | STSW | SSW | SW | S3W variants |
|---|---|---|---|---|
| $d = 9$ | $\lambda = 10.0$ | $\lambda = 20.0$ | $\lambda = 1.0$ | $\lambda = 0.5$ |
| $d = 2$ | $\lambda = 10.0$ | $\lambda = 20.0$ | $\lambda = 1.0$ | $\lambda = 0.1$ |

### B.7 GENERATIVE MODELS

**Setup.** We use Adam (Kinga et al., 2015) optimizer with learning rate $lr = 10^{-3}$. We train with a batch size of 500 over 100 epochs using BCE loss as our reconstruction loss. We choose $L = 200$ trees and $k = 10$ lines for STSW. Following the same settings in Tran et al. (2024b), we fix $L = 100$ projections for others, $N_R = 5$ rotations for RI-S3W and ARI-S3W, and a pool size of 100 random rotations ARI-S3W. We use prior 10 vMFs, $\lambda = 1$ for STSW, $\lambda = 10$ for SSW, and $\lambda = 10^{-3}$ for SW and S3W variants.

**Additional Results on MNIST.** For quantitative analysis, we train the SWAE framework on MNIST and report the FID score in Table 8, along with the generated images in Figure 14. We follow the same settings as in Tran et al. (2024b), which use the latent prior $\mathcal{U}(S^2)$ and train the model with a batch size of 500 over 100 epochs. For STSW, we fix $L = 200$ trees and $k = 10$ lines with a learning rate $LR = 0.01$ and $\lambda = 1$. For other sliced methods, we use $L = 100$ projections and a learning rate $LR = 10^{-3}$, as described in Tran et al. (2024b). The FID scores are computed using 10,000 samples from the test set.

We use the same model architecture as specified in Tran et al. (2024b).

**CIFAR-10 Model Architecture.**

Encoder:

$$x \in \mathbb{R}^{3 \times 32 \times 32} \rightarrow \text{Conv2d}_{32} \rightarrow \text{ReLU} \rightarrow \text{Conv2d}_{32} \rightarrow \text{ReLU}$$
$$\rightarrow \text{Conv2d}_{64} \rightarrow \text{ReLU} \rightarrow \text{Conv2d}_{64} \rightarrow \text{ReLU}$$
$$\rightarrow \text{Conv2d}_{128} \rightarrow \text{ReLU} \rightarrow \text{Conv2d}_{128} \rightarrow \text{Flatten}$$
$$\rightarrow \text{FC}_{512} \rightarrow \text{ReLU} \rightarrow \text{FC}_3$$
$$\rightarrow \ell^2 \text{ normalization} \rightarrow z \in \mathbb{S}^2$$

Decoder:

$$z \in \mathbb{S}^2 \rightarrow \text{FC}_{512} \rightarrow \text{FC}_{2048} \rightarrow \text{ReLU}$$

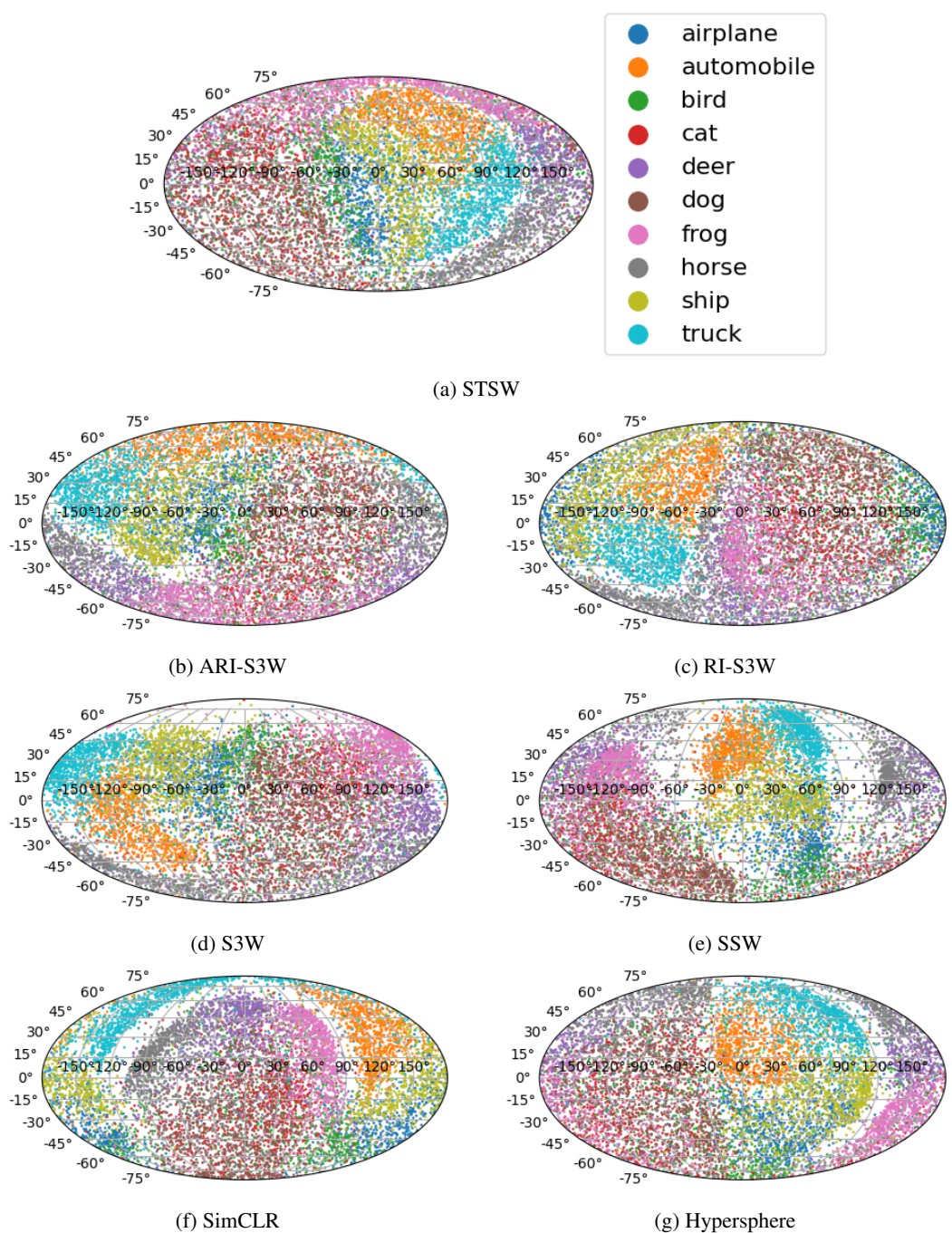

Figure 12: Distributions of CIFAR-10 validation set on $\mathbb{S}^2$ after pre-training.

$$\rightarrow \text{Reshape}(128 \times 4 \times 4) \rightarrow \text{Conv2dT}_{128} \rightarrow \text{ReLU}$$
$$\rightarrow \text{Conv2dT}_{64} \rightarrow \text{ReLU} \rightarrow \text{Conv2dT}_{64} \rightarrow \text{ReLU}$$
$$\rightarrow \text{Conv2dT}_{32} \rightarrow \text{ReLU} \rightarrow \text{Conv2dT}_{32} \rightarrow \text{ReLU}$$
$$\rightarrow \text{Conv2dT}_3 \rightarrow \text{Sigmoid}$$

**MNIST Model Architecture.**

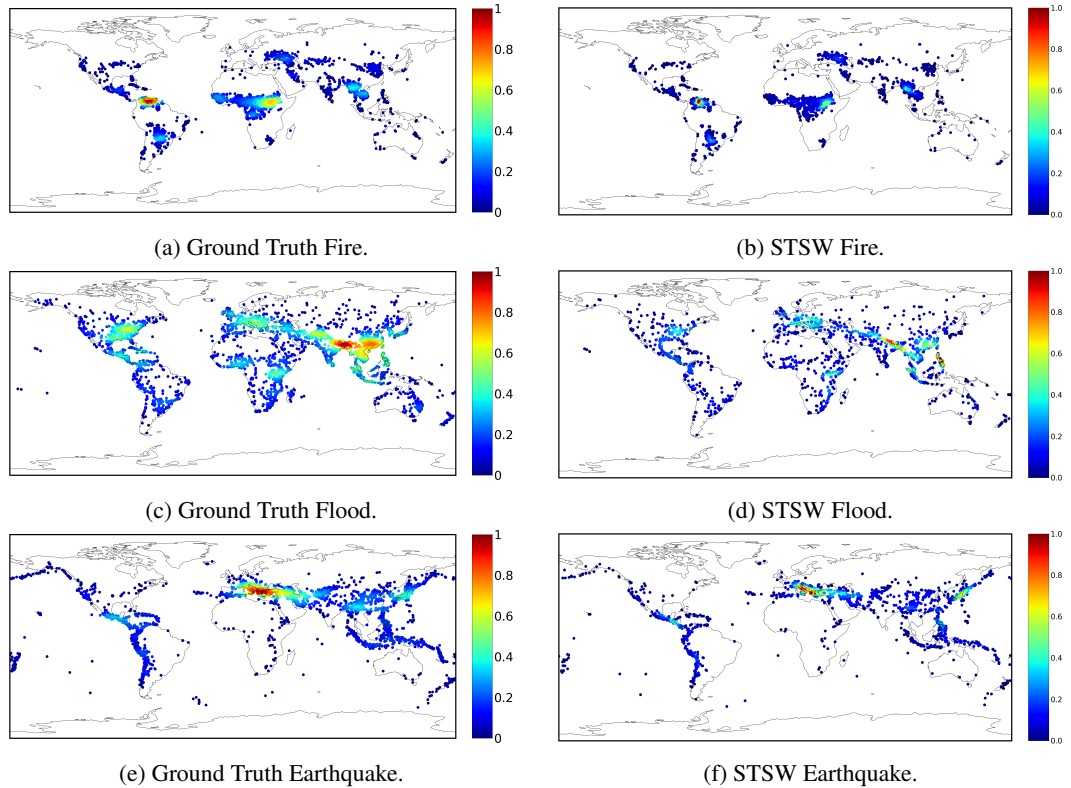

Figure 13: Density estimation on earth data. The left figures (ground truth) represent training data estimated with KDE. The right ones depict the normalized log likelihood of the trained models on test data.

Table 7: Earth datasets.

|  | Earthquake | Flood | Fire |
| --- | --- | --- | --- |
| Train | 4284 | 3412 | 8966 |
| Test | 1836 | 1463 | 3843 |
| Data size | 6120 | 4875 | 12809 |

Encoder:

$$x \in \mathbb{R}^{28 \times 28} \to \text{Conv2d}_{32} \to \text{ReLU} \to \text{Conv2d}_{32} \to \text{ReLU}$$
$$\to \text{Conv2d}_{64} \to \text{ReLU} \to \text{Conv2d}_{64} \to \text{ReLU}$$
$$\to \text{Conv2d}_{128} \to \text{ReLU} \to \text{Conv2d}_{128} \to \text{Flatten}$$
$$\to \text{FC}_{512} \to \text{ReLU} \to \text{FC}_3$$
$$\to \ell^2 \text{ normalization} \to z \in \mathbb{S}^2$$

Decoder:

$$z \in \mathbb{S}^2 \to \text{FC}_{512} \to \text{FC}_{512} \to \text{ReLU}$$
$$\to \text{Reshape}(128 \times 2 \times 2) \to \text{Conv2dT}_{128} \to \text{ReLU}$$
$$\to \text{Conv2dT}_{64} \to \text{ReLU} \to \text{Conv2dT}_{64} \to \text{ReLU}$$
$$\to \text{Conv2dT}_{32} \to \text{ReLU} \to \text{Conv2dT}_{32} \to \text{ReLU}$$
$$\to \text{Conv2dT}_1 \to \text{Sigmoid}$$

Table 8: Average FID of 5 runs on MNIST.

| Method | FID ↓ |
|---|---|
| SW | $73.35 \pm 2.01$ |
| SSW | $76.14 \pm 2.73$ |
| S3W | $75.55 \pm 2.80$ |
| RI-S3W (10) | $72.80 \pm 3.39$ |
| ARI-S3W (30) | $70.37 \pm 2.58$ |
| STSW | $\mathbf{69.16 \pm 2.74}$ |

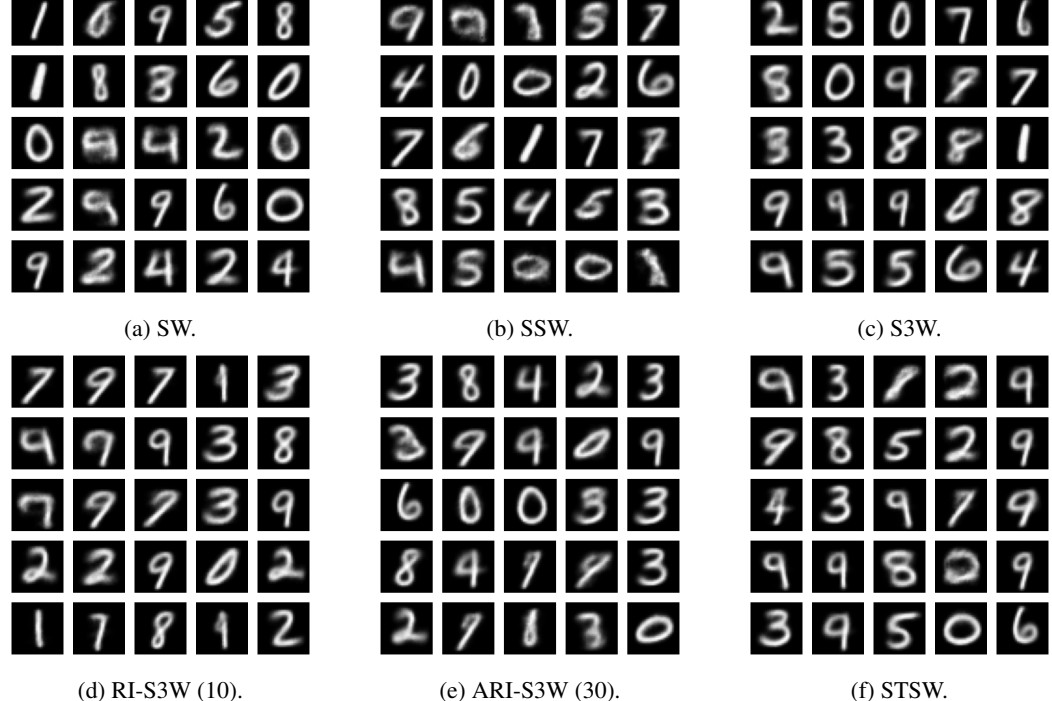

(a) SW.     (b) SSW.     (c) S3W.

(d) RI-S3W (10).     (e) ARI-S3W (30).     (f) STSW.

Figure 14: Generated images of different methods on MNIST of SWAE.

