# OpenReview forum: "Spherical Tree-Sliced Wasserstein Distance"
_ICLR.cc/2025/Conference — ICLR 2025 Poster_

### Official Review · Reviewer_Gf54 · 2024-10-26

**Soundness:** 3
**Presentation:** 3
**Contribution:** 2
**Rating:** 6
**Confidence:** 4

**Summary:**

The paper introduces a novel way to measure distances between probability distributions on hyperspheres, coined Spherical Tree-Sliced Wasserstein (STSW). The core technical contribution lies in adapting tree-based structures from [1] to work on spheres (spherical trees) and defining a new type of Radon transform for these structures. The authors prove this approach leads to closed-form solutions for optimal transport problems and show that STSW is a valid distance metric on $\mathcal{P}(\mathbb{S}^d)$. Through various experiments including gradient flows, density estimation, and self-supervised learning, they show that STSW can perform competitively with/better than the baselines while having faster runtime.

---

[1] Tran, Viet-Hoang, et al. "Tree-Sliced Wasserstein Distance on a System of Lines." arXiv preprint arXiv:2406.13725 (2024).

**Strengths:**

S1. This method addresses an important problem of comparing distributions on the sphere, which has applications in many fields.

S2. Extending the tree-sliced concept to spherical domains is novel and non-trivial.

S3. The writing in the main paper is rigorous, coherent, and easy to follow.

S4. STSW is proved to be a proper metric (compared to SSW [2] which is only known to be pseudometric) with a novel splitting maps that preserve orthogonal invariance.

S5. STSW appears to outperforms various baselines in terms of runtime and other quantitative metrics.

S6. The algorithm is straightforward to implement.

---

[2] Bonet, Clément, et al. "Spherical sliced-wasserstein." arXiv preprint arXiv:2206.08780 (2022).

**Weaknesses:**

W1. Sampling: In the algorithm, the authors propose sampling uniformly from R^{d+1} and then normalizing to get points on S^d, which does not produce a uniform distribution on the sphere. Would this induce a bias (and implications)?

W2. Ablation: The paper would be strengthened if there are more insights provided via ablations on different design choices (i.e., rays, trees). How does the current tree structure help capture the data better than existing methods? Are there limitations, theoretical issues, or numerical instability associated with different components of the method (i.e., S3W [3] has the north pole issue). Can the splitting maps be learned? etc.

W3. Runtime and Complexity: It would be nice to have an explicit discussion of the computational/memory complexity (this is aside from the information provided in Appendix B).

W4. Experiments: This may be a minor point, but setup and hyperparameters could be better documented for all methods. In addition, there is no comparison with Vertical SW in appropriate setups. For generative experiments, there are no samples, of quantitative measures of the quality of images (i.e. the FID score).

---

[3] Tran, Huy, et al. "Stereographic Spherical Sliced Wasserstein Distances." Forty-first International Conference on Machine Learning. 2024.

**Questions:**

- What makes STSW run faster than S3W [3], and in some cases, SW?

- The tree structure is supposed to capture 'richer' topological information per the authors' claim. How does that translate to practical results? Have the authors explored different hyperparameters to confirm that the superior performance in these setups is due to the tree component of the method? Traditional trees in euclidean spaces often have a hierarchical structure; here, it appears that our design choice is not hierarchical? If so, then what are the concrete benefits?

- Are there relationships to the OT distance on the spheres?

---

> ### Author Response · Authors · 2024-11-18
>
> We appreciate the reviewer’s feedback and have provided the following responses to address the concerns raised about our paper. Below, we summarize the weaknesses and questions highlighted by the reviewer and provide our answers accordingly.
>
> ---
>
> **W1. Sampling: In the algorithm, the authors propose sampling uniformly from R^{d+1} and then normalizing to get points on S^d, which does not produce a uniform distribution on the sphere. Would this induce a bias (and implications)?**
>
> **Answer.** This is a typo in our manuscript, and we thank the reviewer to point it out. The correct term is *standard normal distribution on $\mathbb{R}^{d+1}$*, not *uniform distribution on $\mathbb{R}^{d+1}$*. We have revised this in our paper. It is worth noting that, except for the origin, the pushforward of the standard normal distribution on $\mathbb{R}^{d+1}$ via normalization map results in a uniform distribution on $\mathbb{S}^d$.
>
>
> **W2 + Q3. Ablation: The paper would be strengthened if there are more insights provided via ablations on different design choices (i.e., rays, trees). How does the current tree structure help capture the data better than existing methods? Are there limitations, theoretical issues, or numerical instability associated with different components of the method (i.e., S3W [3] has the north pole issue). Can the splitting maps be learned? etc.**
>
> **Are there relationships to the OT distance on the spheres?**
>
> **Answer.** We provided some ablation studies of STSW in Appendix B. Given that the paper focuses on the construction of spherical trees and the corresponding Radon Transform, and the content is already comprehensive, we have decided to leave the analysis of tree structures, and statistical aspects of STSW for future work.
>
> It is worth noting that analyzing these aspects of STSW is challenging due to the introduction of splitting maps. This component is unique to Tree-Sliced Wasserstein variants, distinguishing them from Sliced Wasserstein variants. We are actively working on analyzing splitting maps, and it appears to be a highly promising research direction.
>
> "Can the splitting maps be learned?": In our paper, the splitting map is designed based on the distance from a point to the edges of spherical trees. Intuitively, the proportion of mass at a given point is proportional to its distance from the tree's edges. Splitting maps could potentially be made learnable by parameterizing them as a multi-layer perceptron (MLP) with a softmax layer at the end, allowing for end-to-end training with the model. However, this is a preliminary idea, and we have not empirically verified it yet. We leave this exciting idea for future work.
>
> **W3 + Q1.** **Runtime and Complexity: It would be nice to have an explicit discussion of the computational/memory complexity (this is aside from the information provided in Appendix B).**
>
> **What makes STSW run faster than S3W [3], and in some cases, SW?**
>
> **Answer.** To demonstrate the scalability of the method, we present the computational complexity of calculating STSW using the closed-form approximation described in Eq. (19). The complexity is $\mathcal{O}(LNlogN + LkNd)$, which is theoretically equivalent to that of many sliced methods, such as SW. In practice, as shown in the Experimental Results section, STSW achieves favorable runtime performance. Notably, the closed-form approximation is a crucial factor contributing to the efficiency of our method.
>
> **W4.** **Experiments: This may be a minor point, but setup and hyperparameters could be better documented for all methods. In addition, there is no comparison with Vertical SW in appropriate setups. For generative experiments, there are no samples, of quantitative measures of the quality of images (i.e. the FID score).**
>
> **Answer.** Answering this question from the reviewer involves preparing code and conducting several experiments. We will provide a detailed response within a few days.

---

> ### Author Response · Authors · 2024-11-18
>
> **Q2.** **The tree structure is supposed to capture 'richer' topological information per the authors' claim. How does that translate to practical results? Have the authors explored different hyperparameters to confirm that the superior performance in these setups is due to the tree component of the method? Traditional trees in euclidean spaces often have a hierarchical structure; here, it appears that our design choice is not hierarchical? If so, then what are the concrete benefits?**
>
> **Answer.** Let us answer these questions by discussing more about the motivation for this paper. It arose from a simple yet intriguing idea: In the framework of Sliced Wasserstein (SW), a probability distribution on $\mathbb{R}^d$ is pushed forward onto a line. This raises the question: what does the resulting distribution reveal about the original one? It is evident that distinct distributions, when projected onto the same line, can become indistinguishable.
>
> The situation is similar in the spherical setting. Given, for example, vertical SW, where each slice corresponds to a great semicircle, after rotating a spherical distribution around the diameter of a slice, the projected distribution of on that slice remains unchanged. This means that distinct distributions can become indistinguishable when projected onto the same slice. However, with spherical trees that include more than one great semicircle, the splitting map comes into play. It allows for differentiating distributions that are otherwise indistinguishable under rotation. As a result, two distributions that vertical SW cannot distinguish due to rotational symmetry can now be separated using the spherical tree structure in STSW.
>
> In summary, with the same number of edges (as vertical SW), and thus the same computational cost, spherical trees in STSW provide a significantly deeper understanding of probability distributions compared to individual edges as in vertical SW. While more complex and hierarchical tree structures could be explored with the potential for improved performance, we opted for the simple structure described in the paper to ensure efficient implementation.
>
> A natural question arises: if a better representation space is desired, why not replace one-dimensional manifold with higher-dimensional manifolds? The answer lies in computational feasibility. Optimal Transport in $\mathbb{R}^d$ for $d>1$ is computationally prohibitive due to the lack of a closed-form solution. In contrast, both vertical SW and STSW offer efficient closed-form expressions, making them more practical.
>
> We believe this intuitive interpretation for STSW adequately addresses the reviewer's concerns.
>
> ----
>
> We sincerely thank the reviewer for the valuable feedback. The typos highlighted by the reviewer have been corrected in the revision of our paper. If our responses satisfactorily address the concerns raised, we kindly hope the reviewer will consider increasing the score of our paper.

---

### Official Review · Reviewer_mQXD · 2024-10-31

**Soundness:** 2
**Presentation:** 3
**Contribution:** 2
**Rating:** 6
**Confidence:** 3

**Summary:**

The paper proposes a variant of the Wasserstein distance for distributions defined on the hypersphere. The proposed metric, called the Spherical Tree-Sliced Wasserstein (STSW) distance, adapts the Tree-Sliced Wasserstein distance to be applicable to the hypersphere by defining a novel spherical Radon transform. The proposed metric is invariant to orthogonal transformations, and can be computed efficiently.

**Strengths:**

* The paper is well-written.
* The paper proposes an efficient extension of the Sliced Wasserstein distances applicable to hyperspheres and the construction of the Radon transform is well-motivated.
* The authors also show desirable properties of their proposal, such as invariance to orthogonal transforms.
* On the practical side, the authors propose an efficient way to compute such a metric and show its effectiveness and superior performance in various experiments.

**Weaknesses:**

* The proposed metric is limited to hyperspheres. The paper's contributions appear to be incremental, primarily combining previously established concepts (tree systems and Sliced Wasserstein distances) and extending them to the sphere setting.
* One of the claimed contributions is a bit misleading. Specifically, the authors claim to derive a closed-form expression; however, its computation still relies on approximations: first, due to the need for sampling trees, and second, by considering only discrete distributions in the explanation of how it is computed. This should be clarified, as the current claim gives the impression that the distance can be computed exactly due to the closed-form expression.
* Line 65 states that the use of tree systems enhances the capture of topological information. However, it is not so clear to me why is this the case. An experiment demonstrating this advantage would be useful.

**Questions:**

- **Impact uniformity loss**: How should we understand $STSW(z^A,\nu)$ in eq. 20?  $z^A$ is a single point, so I assume we are considering a Dirac distribution at $z^A$. Theoretically, the distance from a Dirac distribution to a uniform distribution in the sphere is constant regardless of where the Dirac distribution is placed, due to invariance to rotations, right?  Why does this term favour uniformity if it is a constant term? Or am I misunderstanding something?
- **Radon Transform measure preservation**: Why does the proposed Radon transformation in eq. 8 transform a probability distribution $\mu$ defined on $\mathbb{S}^d$ into a probability distribution defined on $\mathcal{T}$? This is mentioned in lines 332-333, but in line 268 it says that $||\mathcal{R}_{\mathcal{T}}^{\alpha}f||\leq||f||_1$.  So it does not immediately follow that the Radon transform preserves the measure.
- **STSW Computation on continuous measures**: In section 5 you explain how to compute STSW in practice, but it is assumed that the probability distributions are discrete.Is it possible to get a closed form analogous to that in eq. 19 for non-discrete distributions?
- **Injectivity of the radon transform**: In Theorem 4.3 it is proved that if the splitting map is $\mathcal{O}(d+1)$-invariant, then the spherical Radon transform is invariant. What would be the consequences of using a non-injective spherical Radon transform? What structure might be missed?

**Minor & Typos:**
- l. 170 "be **a** positive"
- l. 372 $\nu(x)=\sum_{j=1}^n$
- In l. 322 change notation of $\delta$ to other letter, in order to avoid possible confusion with the Dirac delta function.

---

> ### Author Response · Authors · 2024-11-18
>
> We appreciate the reviewer’s feedback and have provided the following responses to address the concerns raised about our paper. Below, we summarize the weaknesses and questions highlighted by the reviewer and provide our answers accordingly.
>
> ---
>
> **W1.  The proposed metric is limited to hyperspheres. The paper's contributions appear to be incremental, primarily combining previously established concepts (tree systems and Sliced Wasserstein distances) and extending them to the sphere setting.**
>
> **Answer.** We believe there is a misunderstanding of the contributions of our paper. Please allow us to clear this misunderstanding by clarifying the novelty of our proposed Spherical Tree-Sliced Wasserstein. Our proposed metric is specifically designed for distributions on hyperspheres. The use of spherical distributions is widespread and is discussed in detail in the Introduction section of the paper. Existing studies on spherical variants of sliced Optimal Transport also primarily focus on data on hyperspheres, such as in [1], [2], [3], and others.
>
> In our view, although the paper combines established concepts like tree systems and Sliced Wasserstein distances, the combination is far from straightforward. For instance:
>
> - The tree systems in [4] and our proposed spherical trees are constructed differently, and their corresponding Radon Transforms also differ significantly.
> - The splitting map in our work is more comprehensively designed than those in [4], as it considers both the positional information of points and lines, rather than just lines. It leads to a theorem for injectivity of our Radon Transform (Theorem 4.3), which requires a non-trivial proof.
>
> For these reasons, we firmly believe our work demonstrates sufficient novelty for the conference.
>
> **W2. One of the claimed contributions is a bit misleading. Specifically, the authors claim to derive a closed-form expression; however, its computation still relies on approximations: first, due to the need for sampling trees, and second, by considering only discrete distributions in the explanation of how it is computed. This should be clarified, as the current claim gives the impression that the distance can be computed exactly due to the closed-form expression.**
>
> **Answer.** We agree with the reviewer that while Eq. (19) provides a closed-form expression for the Wasserstein distance in a tree metric, it does not yield a closed-form expression for STSW, but rather an approximation. Nonetheless, this approximation facilitates efficient implementation, and empirical results demonstrate that STSW performs effectively with this approach.
>
> We have revised our paper according to the reviewer's feedback.
>
> **W3. Line 65 states that the use of tree systems enhances the capture of topological information. However, it is not so clear to me why is this the case. An experiment demonstrating this advantage would be useful.**
>
> **Answer.** Please allow us to clarify the motivation for our paper. It arose from a simple yet intriguing idea: In the framework of Sliced Wasserstein (SW), a probability distribution on $\mathbb{R}^d$ is pushed forward onto a line. This raises the question: what does the resulting distribution reveal about the original one? It is evident that distinct distributions, when projected onto the same line, can become indistinguishable.
>
> The situation is similar in the spherical setting. Given, for example, vertical SW, where each slice corresponds to a great semicircle, after rotating a spherical distribution around the diameter of a slice, the projected distribution of on that slice remains unchanged. This means that distinct distributions can become indistinguishable when projected onto the same slice. However, with spherical trees that include more than one great semicircle, the splitting map comes into play. It allows for differentiating distributions that are otherwise indistinguishable under rotation. As a result, two distributions that vertical SW cannot distinguish due to rotational symmetry can now be separated using the spherical tree structure in STSW.
>
> In summary, with the same number of edges (as vertical SW), and thus the same computational cost, spherical trees in STSW provide a significantly deeper understanding of probability distributions compared to individual edges as in vertical SW.
>
> A natural question arises: if a better representation space is desired, why not replace one-dimensional manifold with higher-dimensional manifolds? The answer lies in computational feasibility. Optimal Transport in $\mathbb{R}^d$ for $d>1$ is computationally prohibitive due to the lack of a closed-form solution. In contrast, both vertical SW and STSW offer efficient closed-form expressions, making them more practical.
>
> We believe this explanation adequately addresses the reviewer's concerns.

---

> ### Author Response · Authors · 2024-11-18
>
> **Q1.** **Impact uniformity loss**: **How should we understand $STSW(z^{A}, v)$ in eq. 20? $z^{A}$ is a single point, so I assume we are considering a Dirac distribution at $z^{A}$. Theoretically, the distance from a Dirac distribution to a uniform distribution in the sphere is constant regardless of where the Dirac distribution is placed, due to invariance to rotations, right? Why does this term favour uniformity if it is a constant term? Or am I misunderstanding something?**
>
> **Answer.** $z^{A}, z^B \in \mathbb{R}^{n \times (d + 1)}$ are the representations from the network projected on the hypersphere of two augmented versions of the same image. Thus, $z^A$ is not a single point. The self-supervised loss in Eq. (20) is proposed in [8] (See Eq. (86)), and we closely follow their setting for this experiment.
>
> **Q2.** **Radon Transform measure preservation**: **Why does the proposed Radon transformation in eq. 8 transform a probability distribution $\mu$ defined on $\mathbb{S}^d$ into a probability distribution defined on $\mathbb{S}^d$ This is mentioned in lines 332-333, but in line 268 it says that $\|| \mathcal{R}^\alpha_\mathcal{T}f \||_{\mathcal{T}} \le \|| f \||_1$. So it does not immediately follow that the Radon transform preserves the measure.**
>
> **Answer.**
> In the case where $f$ is a probability distribution on $\mathbb{S}^d$, $\mathcal{R}^\alpha_\mathcal{T}f$ is a distribution of $\mathcal{T}$. This follows directly from the proof provided in Appendix A.1. We recall the proof as follows: Since $f$ is non-negative on $\mathbb{S}^d$, $\mathcal{R}^\alpha_\mathcal{T}f$ is also non-negative on $\mathcal{T}$. Moreover,
>
> $\|\|\mathcal{R}^\alpha_{\mathcal{T}}f\|\|_{\mathcal{T}}$
>
> $= \sum_{i=1}^k \int_{0}^\pi \left|\mathcal{R}^\alpha_{\mathcal{T}}f(t, r^x_{y_i}) \right|  \, dt$
>
> $= \sum_{i=1}^k \int_{0}^\pi \left|\int_{\mathbb{S}^d} f(y) \cdot \alpha(y, \mathcal{T})_i \cdot \delta(t - \operatorname{arccos}\left<x,y \right>) ~ dy \right| ~ dt$
>
> $= \sum_{i=1}^k \int_{0}^\pi \left(\int_{\mathbb{S}^d} |f(y)| \cdot \alpha(y, \mathcal{T})_i \cdot \delta(t - \operatorname{arccos}\left<x,y \right>) ~ dy \right) ~ dt$
>
> $= \sum_{i=1}^k \int_{\mathbb{S}^d} \left(\int_{0}^\pi  |f(y)| \cdot \alpha(y, \mathcal{T})_i \cdot \delta(t - \operatorname{arccos}\left<x,y \right>) ~ dy \right) ~ dt$
>
> $= \sum_{i=1}^k \int_{\mathbb{S}^d} |f(y)| \cdot \alpha(y, \mathcal{T})\_i \cdot \left( \int_{0}^\pi  \delta(t - \operatorname{arccos}\left<x,y \right>) ~ dt \right) ~ dy$
>
> $= \sum_{i=1}^k \int_{\mathbb{S}^d} |f(y)| \cdot \alpha(y, \mathcal{T})_i ~ dy$
>
> $= \int_{\mathbb{S}^d} |f(y)| \cdot \left (\sum_{i=1}^k \alpha(y, \mathcal{T})_i \right) ~ dy$
>
> $= \int_{\mathbb{S}^d} |f(y)| ~ dy$
>
> $= \||f\||_1 =1.$
>
>
> Thus, $\mathcal{R}^\alpha_\mathcal{T}f$ is a probability distribution on $\mathcal{T}$.
>
> **Q3.** **STSW Computation on continuous measures**: **In section 5 you explain how to compute STSW in practice, but it is assumed that the probability distributions are discrete. Is it possible to get a closed form analogous to that in eq. 19 for non-discrete distributions?**
>
> **Answer.** Equation (19) is derived directly from the closed-form expression presented in [6]. For a general probability distribution, a closed-form expression can be obtained by replacing the summation with integration, as demonstrated in [7]. This approach is analogous to the well-known closed-form expression for the 1-dimensional Wasserstein distance: formulas for general distributions involve integrations, while those for discrete distributions use summation. In practice, implementations for discrete distributions rely on fundamental operations such as matrix multiplication, sorting, and similar techniques.
>
> In applications, we typically work with discrete probability distributions. This is why we focus on discrete probabilities in the paper.
>
>
> **Q4.** **Injectivity of the radon transform**: **In Theorem 4.3 it is proved that if the splitting map is invariant, then the spherical Radon transform is invariant. What would be the consequences of using a non-injective spherical Radon transform? What structure might be missed?**
>
> **Answer.** This is a significant contribution of our paper, as the injectivity of a Radon transform variant is often a crucial requirement. It determines whether the derived metric (such as STSW) qualifies as a true metric or remains a pseudo-metric. Without injectivity, the Radon transform could lead to a pseudo-metric, allowing the possibility of two distinct probability distributions having a distance of zero. Consequently, using a pseudo-metric in applications could result in unstable performance.
>
> The study of injectivity in Radon Transform variants has been extensively explored in numerous studies, including [1], [2], [3], [4], [5], and others.

---

> ### Author Response · Authors · 2024-11-18
>
> **Reference.**
>
> [1] Tran, Huy, et al. "Stereographic spherical sliced wasserstein distances." arXiv preprint arXiv:2402.02345 (2024).
>
> [2] Quellmalz, Michael, Robert Beinert, and Gabriele Steidl. "Sliced optimal transport on the sphere." Inverse Problems 39.10 (2023):105005.
>
> [3] Quellmalz, Michael, Léo Buecher, and Gabriele Steidl. "Parallelly sliced optimal transport on spheres and on the rotation group." Journal of Mathematical Imaging and Vision (2024): 1-26.
>
> [4] Tran, Viet-Hoang, et al. "Tree-Sliced Wasserstein Distance on a System of Lines." arXiv preprint arXiv:2406.13725 (2024).
>
> [5] Kolouri, Soheil, et al. "Generalized sliced wasserstein distances." Advances in neural information processing systems 32 (2019).
>
> [6] Le, Tam, et al. "Tree-sliced variants of Wasserstein distances." Advances in neural information processing systems 32 (2019).
>
> [7] Le, Tam, et al. "Sobolev transport: A scalable metric for probability measures with graph metrics." International Conference on Artificial Intelligence and Statistics. PMLR, 2022.
>
> [8] Bonet, Clément, et al. "Spherical sliced-wasserstein." arXiv preprint arXiv:2206.08780 (2022).
>
> ---
>
> We sincerely thank the reviewer for the valuable feedback. The typos highlighted by the reviewer have been corrected in our paper, and we plan to update the manuscript within a few days.
>
> If our responses satisfactorily address all the concerns raised, we kindly hope the reviewer will consider increasing the score of our paper.

---

> > ### Comment · Reviewer_mQXD · 2024-11-21
> >
> > Thank you for your response. All my concerns have been addressed, and it has helped clarify some misunderstandings. I will raise my score to 6.
> >
> > Regarding the enhancement of topological information using tree systems, while your explanation is clear and reasonable, including a toy example to illustrate this advantage could be a beneficial addition.

---

> > > ### Author Response · Authors · 2024-11-21
> > > **Thanks for your endorsement!**
> > >
> > > Thanks for your response and an additional suggestion. We appreciate your endorsement and will think of a toy example that illustrates the enhancement of topological information using tree systems to include in our revision.

---

### Official Review · Reviewer_k7Wt · 2024-11-03

**Soundness:** 3
**Presentation:** 3
**Contribution:** 3
**Rating:** 8
**Confidence:** 3

**Summary:**

This paper introduces the Spherical Tree-Sliced Wasserstein (STSW) distance, a novel metric designed for optimal transport on spherical domains. The key innovation lies in the integration over spherical trees as the domain, rather than traditional one-dimensional lines or great semicircles used in existing spherical Sliced Wasserstein approaches. This change allows STSW to better capture the underlying topology of spherical data, offering closed-form solutions that enhance both performance and computational efficiency.

The authors introduce a variant of the spherical Radon Transform tailored for spherical trees and prove its injectivity. Defining the STSW in terms of this transform is essential for establishing the metric properties of the distance, including its invariance under the action of the orthogonal group.

**Strengths:**

The paper is well-structured, presenting clear objectives and a comprehensive review of related work.
The efficiency of the new metric is well presented in the experiments.

Leveraging on the ideas presented in Tran et al. (2024b), as said before, this reviewer things that the key innovation of this article (over articles as Bonet et al. (2022) and Tran et al. (2024a)) lies in the integration over spherical trees for defining the new metric between spherical probability measures.

**Weaknesses:**

The approach builds on previous work by Bonet et al. (2022) and Tran et al. (2024a), in the sense that uses the same hight-level ideas.
However, while the research incrementally follows the line of previous studies by Bonet et al. (2022), Tran et al. (2024a), and Tran et al. (2024b), it offers meaningful advancements by developing a metric specifically adapted for spherical data analysis.
Also, the experiments closely follows experiments previously presnted in papers as Bonet et al. (2022).

**Questions:**

Besides the experimental comparisons of the new STSW with SW, SSW, and S3W variants, are there any analytic comparisons among them? Which are the differents in the topologies defined by these different approach?

---

> ### Author Response · Authors · 2024-11-18
>
> We appreciate the reviewer’s feedback and have provided the following responses to address the concerns raised about our paper. Below, we summarize the weaknesses and questions highlighted by the reviewer and provide our answers accordingly.
>
> ---
>
> **W1. The approach builds on previous work by Bonet et al. (2022) and Tran et al. (2024a), in the sense that uses the same hight-level ideas. However, while the research incrementally follows the line of previous studies by Bonet et al. (2022), Tran et al. (2024a), and Tran et al. (2024b), it offers meaningful advancements by developing a metric specifically adapted for spherical data analysis. Also, the experiments closely follows experiments previously presnted in papers as Bonet et al. (2022).**
>
> **Answer.** This is an interesting point and we are enthusiastic about delving deeper into it. Roughly speaking, the tree-sliced framework that is adapted in our paper is built on two key insights:
>
> - Local perspective: Each edge in a spherical tree is treated similarly to an one-dimensional slice in existing Spherical Sliced Wasserstein (SSW) frameworks. Splitting maps determine how the mass at each point is distributed across the lines, and then the projection of these mass portions onto the lines is processed in the same way as in SW.
> - Global Perspective: Spherical tree structures and splitting maps establish connections between the lines, creating a cohesive system. This introduces a novel aspect compared to SSW, enabling interaction and integration among the edges in a spherical tree. The Wasserstein distance can now be computed on this space with a closed-form expression, analogous to how one-dimensional manifolds are treated in SSW.
>
> It is important to note that while these ideas might seem straightforward, their development is non-trivial. A  key challenge lies in ensuring the injectivity of the corresponding Radon Transform, which is critical in determining whether the proposed metric is a true metric or merely a pseudo-metric. We have addressed this issue by providing a rigorous proof in the paper.
>
> In the experimental sections, we follow the standard experimental setups described in [1] and [2], which are widely used benchmarks for assessing the performance of Spherical Sliced Wasserstein methods.
>
> **Q1. Besides the experimental comparisons of the new STSW with SW, SSW, and S3W variants, are there any analytic comparisons among them? Which are the differents in the topologies defined by these different approach?**
>
> **Answer.** Given that the paper focuses on the construction of spherical trees and the corresponding Radon Transform, and the content is already comprehensive, we have decided to leave the analytical and statistical aspects of STSW for future work.
>
> It is worth noting that analyzing these aspects of STSW is challenging due to the introduction of splitting maps. This component is unique to Tree-Sliced Wasserstein variants, distinguishing them from Sliced Wasserstein variants. We are actively working on analyzing splitting maps, and it appears to be a highly promising research direction.
>
> ---
>
> **Reference.**
>
> [1] Bonet, Clément, et al. "Spherical sliced-wasserstein." arXiv preprint arXiv:2206.08780 (2022).
>
> [2] Tran, Huy, et al. "Stereographic spherical sliced wasserstein distances." arXiv preprint arXiv:2402.02345 (2024).
>
> ---
>
> We sincerely thank the reviewer for the valuable feedback. If our responses adequately address all the concerns raised, we kindly hope the reviewer will consider raising the score of our paper.

---

> ### Author Response · Authors · 2024-11-23
> **Any Questions from Reviewer k7Wt on Our Rebuttal?**
>
> We would like to thank the reviewer again for your thoughtful reviews and valuable feedback.
>
> We would appreciate it if you could let us know if our responses have addressed your concerns and whether you still have any other questions about our rebuttal. We would be happy to do any follow-up discussion or address any additional comments.
>
> If you agree that our responses to your reviews have addressed the concerns you listed, we kindly ask that you consider whether raising your score would more accurately reflect your updated evaluation of our paper. Thank you again for your time and thoughtful comments!

---

> ### Comment · Reviewer_k7Wt · 2024-11-24
>
> I thank the authors for clarifying some of my queries.
> Particularly, I agree with their comment "It is important to note that while these ideas might seem straightforward, their development is non-trivial. **A key challenge lies in ensuring the injectivity of the corresponding Radon Transform, which is critical in determining whether the proposed metric is a true metric or merely a pseudo-metric. We have addressed this issue by providing a rigorous proof in the paper.**"
> I will increase my score. I have no further comments.

---

> > ### Author Response · Authors · 2024-11-24
> > **Thanks for your endorsement!**
> >
> > Thanks for your response. We appreciate your endorsement and your acknowledgment of our contributions.

---

### Official Review · Reviewer_8AUo · 2024-11-04

**Soundness:** 3
**Presentation:** 3
**Contribution:** 3
**Rating:** 6
**Confidence:** 3

**Summary:**

This paper introduces the Spherical Tree-Sliced Wasserstein (STSW) distance, an efficient optimal transport (OT) metric for measures on a spheres. By leveraging a novel spherical Radon transform that integrates over spherical tree structures, it provides closed-form OT solutions and maintains computational efficiency. Theoretical analysis and experiments, including gradient flows and self-supervised learning, confirm STSW’s effectiveness and its performance against recent benchmarks.

**Strengths:**

* Paper is very well-written.
* Through extensive experiments, the effectiveness of the STSW has been investigated.
* Paper introduces a novel Radon transform for the measures on the spherical trees.

**Weaknesses:**

* While STSW aims to be efficient, the spherical Radon transform and tree-slicing require considerable computation, especially as the number of edges or the dimension of the hypersphere increases. This could limit scalability for very high-dimensional or densely-sampled spherical data, impacting runtime in large-scale applications. Although I understand that the authors have provided extensive runtime comparisons, I would like to see the scalability of the method on higher-dimensional tasks beyond CIFAR, MNIST, and similar datasets.

* The effectiveness of STSW relies on the quality of sampled spherical trees, which introduces variability in metric accuracy. If the sampling process fails to capture diverse spherical structures adequately, STSW’s results might be inconsistent, especially in complex distributions where more refined tree structures are necessary. I would like to see which strategies (e.g., Markov Chains, Random Paths, etc.) could be applied here to sample more informative slices.

**Questions:**

* I was wondering if incorporating a Markov chain over the distributions of the slices, instead of using a uniform distribution, could help in generating more informative tree-slices.

* I am interested in understanding the topology of the tree corresponding to the most informative slice in spherical trees and compare its effectiveness to the most informative slice in SSW, and essentially comparing them to the most informative slice in SWD. (By the most informative slice I mean Max-slice). This can be done on a chosen benchmark.

---

> ### Author Response · Authors · 2024-11-18
>
> We appreciate the reviewer’s feedback and have provided the following responses to address the concerns raised about our paper. Below, we summarize the weaknesses and questions highlighted by the reviewer and provide our answers accordingly.
>
> ---
>
> **W1. While STSW aims to be efficient, the spherical Radon transform and tree-slicing require considerable computation, especially as the number of edges or the dimension of the hypersphere increases. This could limit scalability for very high-dimensional or densely-sampled spherical data, impacting runtime in large-scale applications. Although I understand that the authors have provided extensive runtime comparisons, I would like to see the scalability of the method on higher-dimensional tasks beyond CIFAR, MNIST, and similar datasets.**
>
> **Answer.** To demonstrate the scalability of the method, we present the computational complexity of calculating STSW using the closed-form approximation described in Eq. (19). The complexity is $\mathcal{O}(LNlogN + LkNd)$, which is theoretically equivalent to that of many sliced methods, such as SW. In practice, as evidenced in the Experimental Results section, STSW exhibits favorable runtime performance. Notably, the closed-form approximation is a crucial factor contributing to the efficiency of our method.
>
> **W2+Q1. The effectiveness of STSW relies on the quality of sampled spherical trees, which introduces variability in metric accuracy. If the sampling process fails to capture diverse spherical structures adequately, STSW’s results might be inconsistent, especially in complex distributions where more refined tree structures are necessary. I would like to see which strategies (e.g., Markov Chains, Random Paths, etc.) could be applied here to sample more informative slices.**
>
>  I was wondering if incorporating a Markov chain over the distributions of the slices, instead of using a uniform distribution, could help in generating more informative tree-slices.
>
> **Answer.** We appreciate the reviewer for suggesting an approach to enhance the STSW method. Exploring more complex distributions for the slices, rather than relying solely on a uniform distribution, is indeed a promising direction. Similar strategies have been adopted in some studies on the Sliced Wasserstein method, such as [1], [2], and others.
>
> For STSW, employing more complex distributions for the slices, such as integrating a Markov chain over the slice distributions, is anticipated to enhance the method's effectiveness. Sampling the roots and edges of spherical trees can be adapted to the data, potentially through a learnable sampling process.
>
> Given that the paper focuses on the construction of spherical trees and the corresponding Radon Transform, and the content is already comprehensive, we have decided to leave a more in-depth exploration of tree sampling methods for future work.
>
> **Q3. I am interested in understanding the topology of the tree corresponding to the most informative slice in spherical trees and compare its effectiveness to the most informative slice in SSW, and essentially comparing them to the most informative slice in SWD. (By the most informative slice I mean Max-slice). This can be done on a chosen benchmark.**
>
> **Answer.** We have conducted an experiment for most informative slice method including MAX_STSW, MAX_SSW and MAX_SW on gradient flow task aimed at learning target distribution of 12 vMFs. We present in table below the results after training for 1000 epochs with learning rate $LR=0.01$. Each experiment is repeated 10 times.
>
> *Table 1: Learning target distribution 12 vFMs, LR=0.01, 1000 epochs, averaged over 10 runs*
> |        | log $W_2$ $\downarrow$ | NLL $\downarrow$ |
> | ------ |:-------:|:--------:|
> | MAX_STSW|  -3.19 $\pm$ 0.03|   -5007.72 $\pm$ 16.34  |
> | MAX_SSW|  -2.76 $\pm$ 0.02 | -4868.78 $\pm$ 60.51|
> | MAX_SW |  -3.10 $\pm$ 0.06 | -4959.14 $\pm$ 12.22 |
>
> We have also included a figure in the paper to visualize this experiment.
>
> ---
>
> **Reference.**
>
> [1] Deshpande, Ishan, et al. "Max-sliced wasserstein distance and its use for gans." Proceedings of the IEEE/CVF conference on computer vision and pattern recognition. 2019.
>
> [2] Nguyen, Khai, et al. "Distributional sliced-Wasserstein and applications to generative modeling." arXiv preprint arXiv:2002.07367 (2020).
>
> ---
> We sincerely thank the reviewer for the valuable feedback. If our responses adequately address all the concerns raised, we kindly hope the reviewer will consider raising the score of our paper.

---

> ### Author Response · Authors · 2024-11-23
> **Any Questions from Reviewer 8AUo on Our Rebuttal?**
>
> We would like to thank the reviewer again for your thoughtful reviews and valuable feedback.
>
> We would appreciate it if you could let us know if our responses have addressed your concerns and whether you still have any other questions about our rebuttal. We would be happy to do any follow-up discussion or address any additional comments.
>
> If you agree that our responses to your reviews have addressed the concerns you listed, we kindly ask that you consider whether raising your score would more accurately reflect your updated evaluation of our paper. Thank you again for your time and thoughtful comments!

---

> > ### Comment · Reviewer_8AUo · 2024-11-25
> >
> > I would like to thank the authors for their thorough rebuttal and for addressing my concerns. I'm satisfied with the current state of the paper and I believe authors are addressing an interesting problem in OT, hence I keep my score unchanged.

---

> ### Author Response · Authors · 2024-11-25
> **Thanks for your endorsement!**
>
> Thanks for your response. We appreciate your endorsement and your acknowledgment of our contributions.

---

### Official Review · Reviewer_oCQ3 · 2024-11-12

**Soundness:** 3
**Presentation:** 3
**Contribution:** 3
**Rating:** 6
**Confidence:** 4

**Summary:**

The paper is a natural extension of sliced spherical OT to incorporate tree systems. The authors propose a topological space on spheres called spherical trees by connecting spherical rays with a common root. They then adapt Radon transform onto the spheres and slice the trees which is equivalent to slicing the spheres with trees. After showing that spherical tree are metric spaces, the authors followed classic approach of adapting the OT computation in the tree-sliced spheres. They provided comprehensive theoretical and empirical results to support their theories.

**Strengths:**

The motivation is justified and the narrative follows standard ones when substituting simpler structures with more sophisticated variants for solving specific OT problems. Claims have been proved by theoretical results and verified by empirical results.

**Weaknesses:**

While I didn't find major weaknesses in the paper, the authors didn't provide sufficient theoretical explanation in several critical places. The proposed STSW outperforms baselines in all most all metrics. First, why did the variances in Table 1 so small, smaller than one tenth of other methods in most lines, under Monte Carlo sampling? Why did STSW outperforms other baselines in terms of runtime? The theoretical complexity the authors provided cannot explain it. Why didn't the better runtime translate to a similar margin in reducing the training time in Table 2? Another point is that the authors attributes the better performances to "the ability to capture topological information of integration
domain" of STSW but they didn't show a direct connection in the paper. The results from the CIFAR dataset are not that impressive and the visualization didn't help either in explaining them.

**Questions:**

419: "STSW outperforms the baselines in all metrics and achieves faster convergence." Why is that? Is this result theoretically predictable?

466: "We also conduct experiments with $d=2$ to visualize learned representations." Why don't we directly project the learned features in the original image space to a sphere, rather than redoing the experiments on the sphere?

443: The variances from STSW are quite small. What's the explanation of that? Why doing tree-slicing on spheres is more robust (against different sampling?)?

We have seen spherical trees in the following article, although it's for a different application and the construction of the trees is different as well. Is there any connection between this work and that?

Meng, Yu, Yunyi Zhang, Jiaxin Huang, Yu Zhang, Chao Zhang, and Jiawei Han. "Hierarchical topic mining via joint spherical tree and text embedding." In Proceedings of the 26th ACM SIGKDD international conference on knowledge discovery & data mining, pp. 1908-1917. 2020.

---

> ### Author Response · Authors · 2024-11-18
>
> We appreciate the reviewer’s feedback and have provided the following responses to address the concerns raised about our paper.
>
> **Q1. 419: "STSW outperforms the baselines in all metrics and achieves faster convergence." Why is that? Is this result theoretically predictable?**
>
> **Answer.** This is an empirical result for the Gradient Flow task. As shown in Table 1 and Figure 10, STSW outperforms the baselines in all metrics and achieves faster convergence. Providing theoretical explanation for this observation requires a deeper exploration of the analytical and statistical dimensions of STSW, which we are currently pursuing as part of our future work on STSW. (See Q3 also.)
>
> **Q2. 466: "We also conduct experiments with
>  to visualize learned representations." Why don't we directly project the learned features in the original image space to a sphere, rather than redoing the experiments on the sphere?**
>
> **Answer.** We closely adhere to the experimental setups outlined in [1] and [2], which are commonly used as benchmarks for evaluating the performance of Spherical Sliced Wasserstein methods. Directly projecting the learned features onto a sphere may lead to some loss of information. Moreover, [1] and [2] provide the visualization of the projections on $\mathbb{S}^{2}$, so we include similar visualizations for better comparison.
>
> **Q3. 443: The variances from STSW are quite small. What's the explanation of that? Why doing tree-slicing on spheres is more robust (against different sampling?)?**
>
> **Answer.** Given that the paper focuses on the construction of spherical trees and the corresponding Radon Transform, and the content is already comprehensive, we have decided to leave the analytical and statistical aspects of STSW for future work.
>
> It is worth noting that analyzing these two aspects of STSW is challenging due to the introduction of splitting maps. This component is unique to Tree-Sliced Wasserstein variants, distinguishing them from Sliced Wasserstein variants. We are actively working on analyzing splitting maps, and it appears to be a highly promising research direction.
>
> **Q4. We have seen spherical trees in the following article, although it's for a different application and the construction of the trees is different as well. Is there any connection between this work and that?**
>
> Meng, Yu, Yunyi Zhang, Jiaxin Huang, Yu Zhang, Chao Zhang, and Jiawei Han. "Hierarchical topic mining via joint spherical tree and text embedding." In Proceedings of the 26th ACM SIGKDD international conference on knowledge discovery & data mining, pp. 1908-1917. 2020.
>
> **Answer.** We appreciate the reviewer for bringing the referenced paper to our attention. While both that paper and ours use the term "spherical tree," the meanings are distinct. In our view, there is likely no connection between the two works.
>
> ---
>
> **Reference.**
>
> [1] Bonet, Clément, et al. "Spherical sliced-wasserstein." arXiv preprint arXiv:2206.08780 (2022).
>
> [2] Tran, Huy, et al. "Stereographic spherical sliced wasserstein distances." arXiv preprint arXiv:2402.02345 (2024).
>
> ---
>
> Once again, we sincerely thank the reviewer for their feedback. Please let us know if there are any additional concerns or questions from the reviewer regarding our paper.

---

> ### Author Response · Authors · 2024-11-23
> **Any Questions from Reviewer oCQ3 on Our Rebuttal?**
>
> We would like to thank the reviewer again for your thoughtful reviews and valuable feedback.
>
> We would appreciate it if you could let us know if our responses have addressed your concerns and whether you still have any other questions about our rebuttal.
>
> We would be happy to do any follow-up discussion or address any additional comments.

---

> > ### Comment · Reviewer_oCQ3 · 2024-11-24
> >
> > I'm lowering the score from 8 to 6. The authors' response is not sufficient. The paper lacks theoretical prediction (and thus explanation) of the results. This is a borderline paper to me.

---

> > > ### Author Response · Authors · 2024-11-24
> > > **Clarification on the Reviewer's Recent Concerns**
> > >
> > > Thank you for your response. We would like to clarify that when we initially wrote our replies to your questions, the weaknesses section in your review was marked as 'to be filled.' This explains why we were not aware of the new concerns you recently added to the updated weaknesses section during the discussion phase.
> > >
> > > We are currently working on addressing these additional concerns and will provide our reply within 1-2 days.

---

> > > > ### Author Response · Authors · 2024-11-24
> > > >
> > > > Below, we provide detailed responses to address the recent concerns raised by the reviewer.
> > > >
> > > > ---
> > > >
> > > > **W1. First, why did the variances in Table 1 so small, smaller than one tenth of other methods in most lines, under Monte Carlo sampling?**
> > > >
> > > > **Answer.** In this Gradient Flow task, we trained all methods for $500$ epochs, with the results averaged over $10$ runs. We used the same settings and hyperparameters as the baselines, including the number of epochs, as described in [2].
> > > >
> > > > Figure $10$ in the paper shows that STSW begins to converge at around $300$ epochs. From the figure, it is evident that STSW converges significantly earlier than other methods, which contributes to its reported low variance.
> > > >
> > > > **W2. Why did STSW outperforms other baselines in terms of runtime? The theoretical complexity the authors provided cannot explain it. Why didn't the better runtime translate to a similar margin in reducing the training time in Table 2?**
> > > >
> > > > **Answer.**
> > > > All our experiments were conducted on a NVIDIA H100 80G.
> > > >
> > > > > Why did STSW outperforms other baselines in terms of runtime? The theoretical complexity the authors provided cannot explain it.
> > > >
> > > > The time complexity for projecting $N$ samples into a tree system is $O(LNd)$, as the projections on lines within the same tree are similar, meaning we only need to account for one line per tree. Sorting the projected coordinates requires $O(LN\log(N)$. Calculating the distance and applying the softmax function to distribute mass across all lines in each tree has a time complexity of $O(LkNd)$. Computing the tree-sliced distance takes $O(LkN)$. Therefore, the total theoretical complexity is $O(LNlogN + LkNd)$.
> > > >
> > > > We thank the authors of [1] for providing code used in their experiments. The implementations of S3W, SW and experiments are taken from [mint-vu/s3wd](https://github.com/mint-vu/s3wd). In our implementation of STSW, we address the communication bottleneck caused by data movement, which is the current limiting factor for GPU performance, especially when $N$ is large. Two major strategies are employed to reduce data movement:
> > > >
> > > > - We combine the source and target data into a single sorting operation, minimizing redundant computations.
> > > > - In a spherical tree with $k$ lines, projecting and sorting data along these lines are identical. Therefore, we perform this process on a single line instead of repeating it for $k$ lines.
> > > >
> > > > These are the reasons behind your observation regarding the runtime of STSW compared to other methods.
> > > >
> > > > >Why didn't the better runtime translate to a similar margin in reducing the training time in Table 2?
> > > >
> > > > There are notable differences in the hyperparameters used between Task 1 and Task 2. For example:
> > > >
> > > > - In Task 1, ARI-S3W employs $L=1000$ projections and $R=5$ rotations; Whereas,
> > > > - In Task 2, it uses $L=200$ projections and $R=30$ rotations as in the original setup.
> > > >
> > > > Additionally, the amount of training data varies significantly between the two tasks: Task 1 involves $2,400$ samples, whereas Task 2 uses $60,000$ samples.

---

> > > > > ### Comment · Reviewer_oCQ3 · 2024-11-24
> > > > >
> > > > > We combine the source and target data into a single sorting operation, minimizing redundant computations.
> > > > >
> > > > > Can one use the same technique on other sliced OT variants?

---

> > > > > > ### Author Response · Authors · 2024-11-25
> > > > > >
> > > > > > Thanks for your question. We understand the reviewer's inquiry about which factors make the difference in the runtime between our methods and other sliced OT variants, as well as whether they can be applied to other methods to boost the runtime. However, we think the difference results from two main factors whose influences vary between OT variants: *the implementation coefficient (explained in the next paragraph)* and *the task's inherent characteristics*. In our paper, we focus on theoretically designing an algorithm that computes STSW while having nearly the same complexity as SW. Additionally, as another contribution of our work, we developed an optimized and efficient implementation of our method that overcomes the low-efficiency limitation of previous Tree-Sliced Wasserstein methods, achieving a comparable runtime with other sliced OT variants. For further clarification, please allow us to explain these two aforementioned factors below.
> > > > > >
> > > > > > First, even if two methods share the same computational complexity, they may still exhibit different *implementation coefficients*, which can lead to notable differences in runtime between the methods. The implementation coefficient quantifies the performance gap between the empirical runtime of an algorithm and its theoretical complexity, expressed as $t_{empirical} / t_{theoretical}$. For example, a single `for` loop over $n$ elements ($O(n)$), e.g., `for i=1 to n;`, has an implementation coefficient of $1$, while two sequential loops over the same $n$ elements ($O(n)$), e.g., `for i=1 to n;` followed by `for i=n to 1;`, have a coefficient of $2$, resulting in double the runtime. In the real world, the implementation coefficient is hard to assess since it is affected by a lot of factors such as hardware design, source code, compiler, memory access pattern, etc.
> > > > > >
> > > > > > Second, for a given task, the runtime of each method is influenced not only by implementation details but also by the task's inherent characteristics. For instance, in the self-supervised learning task (Table 2 in our manuscript), STSW demonstrates a slightly faster runtime than SW. However, in the Sliced-Wasserstein autoencoder task (Table 4 in our manuscript), STSW exhibits a slower runtime compared to SW. This difference could arise from various factors, such as system hardware configurations (e.g., GPU vs. CPU, presence or absence of cache, bandwidth between components) and differences in data movement, among others.
> > > > > >
> > > > > > Because of the two factors discussed above, it is understandable why our proposed method has nearly the same complexity as SW but obtains a different runtime. To explain this, as we have pointed out, is generally difficult, requiring a comprehensive performance analysis of the code and implementation of each method, and beyond the scope of our paper.
> > > > > >
> > > > > > We hope our responses have resolved your concerns. If you believe that our replies have adequately addressed the issues you raised, we kindly ask you to consider whether updating your score would more accurately reflect your updated evaluation of our paper. Thank you once again for your time and thoughtful feedback!

---

> ### Author Response · Authors · 2024-11-24
>
> **W3. Another point is that the authors attributes the better performances to "the ability to capture topological information of integration domain" of STSW but they didn't show a direct connection in the paper.**
>
> **Answer.** Let us discuss the motivation behind our paper. It comes from a simple yet intriguing idea: In the framework of Sliced Wasserstein (SW), a probability distribution on $\mathbb{R}^d$ is pushed forward onto a line. This raises the question: what does the resulting distribution reveal about the original one? It is evident that distinct distributions, when projected onto the same line, can become indistinguishable.
>
> The situation is similar in the spherical setting. Given, for example, vertical SW, where each slice corresponds to a great semicircle, after rotating a spherical distribution around the diameter of a slice, the projected distribution of on that slice remains unchanged. This means that distinct distributions can become indistinguishable when projected onto the same slice. However, with spherical trees that include more than one great semicircle, the splitting map comes into play. It allows for differentiating distributions that are otherwise indistinguishable under rotation. As a result, two distributions that vertical SW cannot distinguish due to rotational symmetry can now be separated using the spherical tree structure in STSW.
>
> In summary, with the same number of edges (as vertical SW), and thus the same computational cost, spherical trees in STSW provide a significantly deeper understanding of probability distributions compared to individual edges as in vertical SW.
>
> A natural question arises: if a better representation space is desired, why not replace one-dimensional manifold with higher-dimensional manifolds? The answer lies in computational feasibility. Optimal Transport in $\mathbb{R}^d$ for $d>1$ is computationally prohibitive due to the lack of a closed-form solution. In contrast, both vertical SW and STSW offer efficient closed-form expressions, making them more practical.
>
> We believe this explanation sufficiently answers the question raised.  It is also the same concern we included in the General Response.
>
>
>
> **W4. The results from the CIFAR dataset are not that impressive and the visualization didn't help either in explaining them.**
>
> **Answer.** For self-supervised learning task, we recall two properties for evaluating the representation quality, which are Alignment and Uniformity (proposed in [1]):
>
> - Alignment: Ensures that similar samples are assigned similar features.
> - Uniformity: Promotes an even distribution of features across the hypersphere.
>
> In Table 2, we consider the reported accuracy to be a significant improvement. For example, focusing on the accuracy of encoded features (Acc. E), S3W-related methods [2] achieve the best result of $80.08\\%$, with the second-best result from other baselines being SimCLR at $79.97\\%$. In contrast, our method achieves an even higher accuracy of $80.53\\%$.
>
>
> Notably, STSW demonstrates competitive efficiency, achieving the second-best runtime at 9.54 seconds per epoch, closely following SimCLR, which achieves the best runtime at 9.34 seconds per epoch.
>
> In Figure 12, we present the learned representations of various methods. We observe that STSW demonstrates a balanced combination of these two properties.
>
> ---
>
> **Reference.**
>
> [1] Wang, Tongzhou, and Phillip Isola. "Understanding contrastive representation learning through alignment and uniformity on the hypersphere." International conference on machine learning. PMLR, 2020.
>
> [2] Huy Tran et al., Stereographic Spherical Sliced Wasserstein Distances. ICML 2024.
>
> ---
>
> We sincerely thank the reviewer for the valuable feedback. If our responses adequately address all the concerns raised, we kindly hope the reviewer will consider raising the score of our paper.

---

### Author Response · Authors · 2024-11-21
**General Response**

Dear AC and reviewers,

Thanks for your thoughtful reviews and valuable comments, which have helped us improve the paper significantly.

We sincerely thank the reviewers for their valuable feedback and constructive suggestions. We are encouraged by the positive endorsements regarding the following aspects of our work:

1. The paper is well-written, well-structured, and the writing is rigorous, coherent, and easy to follow. (Reviewer Gf54, mQXD, k7Wt, 8AUo, oCQ3)

2.  The idea of proposing an efficient extension of Sliced Wasserstein distances to hyperspheres, with a well-motivated Radon transform is both novel and non-trivial, addressing the important problem of comparing distributions on spheres with applications across various fields; Furthermore, the proposed distance is rigorously shown to be a proper metric with orthogonal invariance. (Reviewer Gf54, mQXD, k7Wt, 8AUo, oCQ3)

3. The proposed distance is shown to be effective and efficient, outperforming baselines in runtime and quantitative metrics through extensive experiments. (Reviewer Gf54, mQXD, k7Wt, 8AUo, oCQ3)

4. The algorithm is straightforward to implement. (Reviewer Gf54)

---

Below, we address a common question raised in the reviews:

**Q1. Tree structure enhances the capture of topological information.**

**Answer.** The motivation for our paper comes from a simple yet intriguing idea: In the framework of Sliced Wasserstein (SW), a probability distribution on $\mathbb{R}^d$ is pushed forward onto a line. This raises the question: what does the resulting distribution reveal about the original one? It is evident that distinct distributions, when projected onto the same line, can become indistinguishable.

The situation is similar in the spherical setting. Given, for example, vertical SW, where each slice corresponds to a great semicircle, after rotating a spherical distribution around the diameter of a slice, the projected distribution of on that slice remains unchanged. This means that distinct distributions can become indistinguishable when projected onto the same slice. However, with spherical trees that include more than one great semicircle, the splitting map comes into play. It allows for differentiating distributions that are otherwise indistinguishable under rotation. As a result, two distributions that vertical SW cannot distinguish due to rotational symmetry can now be separated using the spherical tree structure in STSW.

In summary, with the same number of edges (as vertical SW), and thus the same computational cost, spherical trees in STSW provide a significantly deeper understanding of probability distributions compared to individual edges as in vertical SW.

A natural question arises: if a better representation space is desired, why not replace one-dimensional manifold with higher-dimensional manifolds? The answer lies in computational feasibility. Optimal Transport in $\mathbb{R}^d$ for $d>1$ is computationally prohibitive due to the lack of a closed-form solution. In contrast, both vertical SW and STSW offer efficient closed-form expressions, making them more practical.

We believe this explanation sufficiently answers the question raised.

---

### Author Response · Authors · 2024-11-21
**Summary of Revisions**

Incorporating comments and suggestions from reviewers, as well as some further empirical studies we believe informative, we summarize here the main changes in the revised paper:
- We clarified in **lines 83, 90, 327, 378, 531, 769, 1160** and **1162** that we derive a closed-form **approximation** for STSW (as noted by Reviewer *mQXD*).
- We changed $\delta$ to $\zeta$ to avoid confusion with the Dirac delta function (as suggested by Reviewer *mQXD*) in **lines 322, 1237, 1329, 1371** and **1495**.
- We corrected typos in **lines 170** and **372** (as suggested by Reviewer *mQXD*).
- We corrected the typo $x^{(l)}, y^{(l)}_1, \ldots, y^{(l)}_k \sim \mathcal{U}(\mathbb{R}^{d+1})$ to $x^{(l)}, y^{(l)}\_1, \ldots, y^{(l)}_k \sim \mathcal{N}(0, Id\_{d+1})$ in  **line 392** (as suggested by Reviewer *Gf54*).
- We added an additional experiment for the most informative sliced methods (as suggested by Reviewer *8AUo*), including MAX-STSW, MAX-SSW and MAX-SW in **Appendix B.3**. We report in **Table 5** and **Figure 11** the negative log-likelihood (NLL) and converged log 2-Wasserstein curves.
- We added an additional experiment on generative task using MNIST dataset (based on recommendations of Reviewer *Gf54*) in **Appendix B.6**. We included the FID scores in **Table 8** and generated images produced by the trained models in **Figure 14**.

---

### Meta-Review · Area_Chair_94ig · 2024-12-18

**Metareview:**

This paper introduces the Spherical Tree-Sliced Wasserstein (STSW) distance, a novel metric for optimal transport on spherical domains, which leverages a unique spherical Radon Transform and tree structures to achieve computational efficiency and enhanced topological representation. A key strength of the paper lies in its ability to adapt tree-sliced methods to hyperspherical distributions, offering both theoretical rigor and practical relevance. The reviewers identified several areas for improvement, including the need for deeper theoretical explanations for runtime and convergence performance, clearer connections between tree structures and topological information capture, and additional experiments on scalability to high-dimensional data. However, the authors addressed these concerns during the rebuttal period by clarifying key concepts, refining experimental sections, and demonstrating responsiveness to reviewer feedback. These updates, coupled with the novelty and practicality of the proposed method, make this paper a strong candidate for acceptance, with the potential to advance research in optimal transport on spherical domains.

**Additional Comments On Reviewer Discussion:**

During the rebuttal period, key issues raised by the reviewers (e.g., Gf54, mQXD, and oCQ3) included concerns about the theoretical foundation and clarity of the results. The authors clarified the injectivity of the spherical Radon Transform and addressed computational optimizations, providing additional experiments on sampling strategies and model performance. Despite these efforts, Reviewer oCQ3 remained unconvinced about theoretical completeness, maintaining a borderline evaluation, while others (e.g., k7Wt and 8AUo) acknowledged meaningful improvements and expressed greater confidence in the paper’s merit. The decision to recommend acceptance was based on the overall novelty, alignment with community interest, and the potential for the final version to incorporate suggested improvements effectively.

---

### Decision · Program_Chairs · 2025-01-22

Accept (Poster)